# New Insight into Hybrid Stochastic Gradient Descent: Beyond With-Replacement Sampling and Convexity

**Pan Zhou**[*]  **Xiao-Tong Yuan**[†]  **Jiashi Feng**[*]
[*] Learning & Vision Lab, National University of Singapore, Singapore
[†] B-DAT Lab, Nanjing University of Information Science & Technology, Nanjing, China
pzhou@u.nus.edu    xtyuan@nuist.edu.cn    elefjia@nus.edu.sg

## Abstract

As an incremental-gradient algorithm, the hybrid stochastic gradient descent (HSGD) enjoys merits of both stochastic and full gradient methods for finite-sum problem optimization. However, the existing rate-of-convergence analysis for HSGD is made under with-replacement sampling (WRS) and is restricted to convex problems. It is not clear whether HSGD still carries these advantages under the common practice of without-replacement sampling (WoRS) for non-convex problems. In this paper, we affirmatively answer this open question by showing that under WoRS and for both convex and non-convex problems, it is still possible for HSGD (with constant step-size) to match full gradient descent in rate of convergence, while maintaining comparable sample-size-independent incremental first-order oracle complexity to stochastic gradient descent. For a special class of finite-sum problems with linear prediction models, our convergence results can be further improved in some cases. Extensive numerical results confirm our theoretical affirmation and demonstrate the favorable efficiency of WoRS-based HSGD.

## 1 Introduction

We consider the following *finite-sum minimization problem*:

$$\min_{\boldsymbol{x} \in \boldsymbol{\mathcal{X}}} f(\boldsymbol{x}) := \frac{1}{n} \sum_{i=1}^{n} f_i(\boldsymbol{x}), \tag{1}$$

where each individual $f_i(\boldsymbol{x})$ is $\ell$-smooth and the feasible set $\boldsymbol{\mathcal{X}} \subseteq \mathbb{R}^d$ is convex. In the field of machine learning, formulation (1) encapsulates a large body of optimization problems including least square regression, logistic regression and deep neural networks training, to name a few. Such a problem can be solved by various algorithms, *e.g.* full gradient descent (FGD) [1], stochastic GD (SGD) [2], hybrid SGD [3], SDCA [4] and SVRG [5].

In this paper, we are particularly interested in Hybrid SGD (HSGD) [3, 6, 7] which is an inexact gradient method that iteratively samples an evolving mini-batch of the terms in (1) for gradient estimation. The iteration of HSGD is given by

$$\boldsymbol{x}^{k+1} = \Phi_{\boldsymbol{\mathcal{X}}}\left(\boldsymbol{x}^k - \eta_k \boldsymbol{g}^k\right), \text{with } \boldsymbol{g}^k = \frac{1}{s_k} \sum_{i_k \in \mathcal{S}_k} \nabla f_{i_k}(\boldsymbol{x}^k),$$

where $\Phi_{\boldsymbol{\mathcal{X}}}(\cdot)$ denotes the Euclidean projection onto $\boldsymbol{\mathcal{X}}$, $\eta_k$ is the learning rate, and $\mathcal{S}_k$ denotes the set of the $s_k$ selected samples at the $k$-th iteration. In early iterations, HSGD selects a few samples to compute the full gradient approximately; and along with more iterations, $s_k$ is increased gradually, leading to more accurate full gradient estimation. Such a mechanism allows HSGD to simultaneously enjoy the merits of both SGD and FGD, *i.e.* rapid initial process of SGD and constant learning rate $\eta_k$ without sacrificing the convergence rate of FGD [6].

**Motivation.** Though HSGD has been shown, both in theory and practice, to bridge smoothly the gap between full and stochastic gradient descent methods, its rate-of-convergence analysis remains restrictive in several aspects.

First, the convergence behavior of HSGD under *without-replacement* sampling (WoRS) is not clear. In the existing analysis [6], the stochastic gradient is assumed to be computed under with-replacement sampling (WRS). But for stochastic optimization, it is a more common practice to use WoRS, *i.e.*, to pass the loss functions $f_i(\boldsymbol{x})$ sequentially, after random shuffling, without revisiting any of them [8, 9]. This makes significant discrepancy between the theoretical guarantee and practical implementation. As shown in Figure 1 (a), WoRS tends to provide better performance than WRS in actual implementation.

Second, the convergence behavior of HSGD for non-convex problems is not clear. Prior convergence guarantees on HSGD are limited to convex problems. Bertsekas [3] established lin-

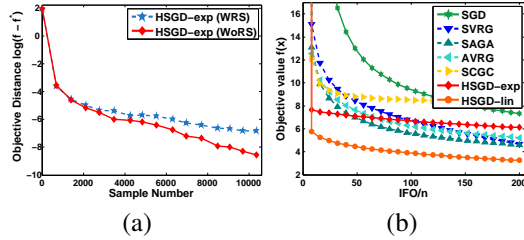

(a)                          (b)

Figure 1: Comparison of WoRS-based HSGD. (a) WoRS *vs.* WRS in HSGD: optimizing a softmax regression model with a single full pass over the data letter. (b) Comparison among randomized algorithms for optimizing a feedforward neural network with 50 full passes over the data sensorless. HSGD-exp and HSGD-lin respectively denote WoRS based HSGD with exponentially and linearly increasing mini-batch sizes (ref. Section 3.2 and 3.4). See more results in supplement.

ear convergence of HSGD for least square problems. Friedlander *et al.* [6] proved that HSGD converges linearly for strongly convex problems with exponentially increasing $s_k$, and sub-linearly for arbitrary convex problems with polynomially increasing $s_k$. Unfortunately, non-convex convergence guarantee on HSGD is still absent, though highly desirable in machine learning applications and extensively studied in other stochastic algorithms, *e.g.* SVRG [10, 11]. In Figure 1 (b), HSGD has sharper convergence behavior than several state-of-the-art SGD methods in training neural networks.

Third, the Incremental First-order Oracle (IFO) complexity (*i.e.* stochastic gradient computation; see Definition 2) of HSGD is largely left unknown. Although Friedlander *et al.* [6] showed that HSGD maintains steady convergence rates of FGD, its IFO complexity is not explicitly analyzed, making it less clear where HSGD should be positioned w.r.t. existing stochastic gradient approaches in overall computational complexity.

**Summary of contributions.** In this work, we address the aforementioned three limitations in the existing analysis of HSGD. We analyze the rate-of-convergence of HSGD under WoRS in a wide problem spectrum including strongly convex, non-strongly convex and non-convex problems. Table 1 summarizes our main results on IFO complexity of HSGD (WoRS) and compares them against state-of-the-art WoRS-oriented results for (stochastic) gradient methods. These results are divided into two groups: for general problems and for a special class of problems with linear prediction loss $f_i(\boldsymbol{x}) = h(\boldsymbol{a}_i^\top \boldsymbol{x})$. As shown in the bottom row of Table 1, we contribute several new theoretical insights into HSGD, which are elaborated in the following paragraphs.

The bounds highlighted in *green*: For both general and certain specially structured strongly convex problems, HSGD is $n\times$ faster than FGD. Compared to the results for SAGA and AVRG [12], the IFO complexity of HSGD is not relying on the sample size $n$ but dependent on $1/\epsilon$. This suggests that HSGD will converge faster when $n$ dominates $1/\epsilon$. Finally, compared to the results for SGD in linear prediction problems [13], ours has removed the dependency on the logarithm term $\log(\kappa/\epsilon)$.

The bounds highlighted in *red*: To our best knowledge, for the first time these new results establish guarantees on WoRS-based stochastic approaches for non-strongly convex and non-convex problems.

The bounds highlighted in *blue*: If the loss function $h(\boldsymbol{a}_i^\top \boldsymbol{x})$ in the linearly structured problem is strongly convex in terms of $\boldsymbol{a}_i^\top \boldsymbol{x}$ (but $f(\boldsymbol{x})$ may still be non-strongly convex), HSGD has $\mathcal{O}(1/\epsilon)$ IFO complexity. The least square regression and logistic regression (with a bounded feasible set) models have such a linear prediction structure.

The bounds highlighted in *brown*: When the specially structured problem is non-strongly convex, HSGD converges to the minimum of problem (1), while SGD can only be shown to converge to a sub-optimum up to some statistical error (see footnote 2 below Table 1).

**Related work.** Understanding randomized algorithms under WoRS and random reshuffling is gaining considerable attention in recent years. By focusing on least squares problems, Recht *et al.* [14] utilized arithmetic-mean inequality on matrices to show that for randomized algorithms, WoRS is always faster than WRS if the data are randomly generated from a certain distribution. For more general

Table 1: Comparison of IFO complexity for randomized algorithms under WoRS. $\kappa = \ell/\rho$ denotes the condition number of $\ell$-smooth and $\rho$-strong convex cases for problem (1). Best viewed in color.

| | General Problem | | | Specially Structured Problem with $f_i(\boldsymbol{x}) = h(\boldsymbol{a}_i^\top \boldsymbol{x})$ | | |
| --- | --- | --- | --- | --- | --- | --- |
| | Stro. conv. | Non-Stro. conv. | Non-conv. | $f(\cdot)$ is stro. conv. | $f(\cdot)$ is non-stro. conv. | $h(\cdot)$ is stro. conv. |
| Metric: $\mathbb{E}\|\boldsymbol{x}^a - \boldsymbol{x}^*\|_2^2 \leq \epsilon$ for stro. conv., $\mathbb{E}[f(\boldsymbol{x}^a) - f(\boldsymbol{x}^*)] \leq \epsilon$ for non-stro. conv., $\mathbb{E}\|\nabla f(\boldsymbol{x}^a)\|_2^2 \leq \epsilon$ for non-conv. | | | | | | |
| FGD [9] | $\mathcal{O}\left(\frac{n\kappa^2}{\epsilon}\right)$ | — | — | $\mathcal{O}\left(\frac{n\kappa^2}{\epsilon}\right)$ | — | — |
| SAGA [12] | $\mathcal{O}\left(n\kappa^2\log\left(\frac{1}{\epsilon}\right)\right)$ | — | — | $\mathcal{O}\left(n\kappa^2\log\left(\frac{1}{\epsilon}\right)\right)$ | — | — |
| AVRG [12] | $\mathcal{O}\left(n\kappa^2\log\left(\frac{1}{\epsilon}\right)\right)$ | — | — | $\mathcal{O}\left(n\kappa^2\log\left(\frac{1}{\epsilon}\right)\right)$ | — | — |
| HSGD | $\mathcal{O}\left(\frac{\kappa^2}{\epsilon}\right)$ | $\mathcal{O}\left(\frac{1}{\epsilon^3}\right)^1$ | $\mathcal{O}\left(\frac{1}{\epsilon^2}\right)$ | $\mathcal{O}\left(\frac{\kappa^2}{\epsilon}\right)$ | $\mathcal{O}\left(\frac{1}{\epsilon^3}\right)$ | $\mathcal{O}\left(\frac{1}{\epsilon}\right)$ |
| Metric: $\mathbb{E}[f(\boldsymbol{x}^a) - f(\boldsymbol{x}^*)] \leq \epsilon$ for both stro. and non-stro. conv., $\mathbb{E}\|\nabla f(\boldsymbol{x}^a)\|_2^2 \leq \epsilon$ for non-conv. | | | | | | |
| SGD [13] | — | — | — | $\mathcal{O}\left(\frac{\kappa}{\epsilon}\log\left(\frac{\kappa}{\epsilon}\right)\right)$ | $\mathcal{O}\left(\frac{1}{\epsilon^2}\right)^2$ | — |
| HSGD | $\mathcal{O}\left(\frac{\kappa}{\epsilon}\right)$ | $\mathcal{O}\left(\frac{1}{\epsilon^3}\right)^1$ | $\mathcal{O}\left(\frac{1}{\epsilon^2}\right)$ | $\mathcal{O}\left(\frac{\kappa}{\epsilon}\right)$ | $\mathcal{O}\left(\frac{1}{\epsilon^3}\right)$ | $\mathcal{O}\left(\frac{1}{\epsilon}\right)$ |

[1] Our IFO complexity for arbitrary convex cases appears higher than the non-convex ones, as we use sub-optimality metric $\mathbb{E}[f(\boldsymbol{x}^a) - f(\boldsymbol{x}^*)] \leq \epsilon$ for convex cases while $\mathbb{E}\|\nabla f(\boldsymbol{x}^a)\|_2^2 \leq \epsilon$ for non-convex cases.

[2] Corollary 1 in [13] provides $\mathbb{E}[f(\boldsymbol{x}^a) - f(\boldsymbol{x}^*)] \leq R_T/k + 2(12 + \sqrt{2}D)/\sqrt{n}$ where $D$ denotes the diameter of the domain $\boldsymbol{\mathcal{X}}$, $k$ is the iteration number and $R_T \sim \mathcal{O}(D\ell/\sqrt{k})$ is the regret bound of SGD for (1). The term $2(12 + \sqrt{2}D)/\sqrt{n}$ is a statistical error which is an artifact from the regret analysis approach.

smooth and strongly convex problems, Gürbüzbalaban *et al.* [9] proved that gradient descent based on random reshuffling enjoys $\mathcal{O}(1/k^2)$ rate of convergence after $k$ epochs, as opposed to $\mathcal{O}(1/k)$ under WRS. But this analysis does not explicitly explain why WoRS works well after a few (or even just one) passes over the data. To answer such a central question, by leveraging regret analysis, Shamir *et al.* [13] proved that for a special class of loss functions $f_i(\boldsymbol{x}) = h(\boldsymbol{a}_i^\top \boldsymbol{x})$, SGD and SVRG using WoRS can achieve competitive IFO complexity to their WRS counterparts. More recently, Ying *et al.* [12] proved that for strongly convex problems, both SAGA [15] and their proposed AVRG algorithm achieve linear convergence rate with WoRS. Recently, Zhou *et al.* [7] applied the HSGD algorithm for solving sparsity or rank-constrained problems and proved its linear convergence rate under the restricted strong convex and smooth conditions. Our work differs from these prior works: 1) For the first time, we provide WoRS based theoretical analysis for HSGD. 2) Our analysis covers non-strongly convex and non-convex cases which are not covered by the current WoRS analysis of stochastic gradient methods.

## 2 Preliminaries

We first introduce the concepts of strong convexity and Lipschtiz smoothness which are commonly used in analyzing stochastic gradient methods [4, 5, 16, 17, 18, 19].

**Definition 1** (Strong convexity and Lipschitz smoothness). *We say a function $g(\boldsymbol{x})$ is $\rho$-strongly-convex if there exists a positive constant $\rho$ such that $\forall \boldsymbol{x}_1, \boldsymbol{x}_2 \in \boldsymbol{\mathcal{X}}$, $g(\boldsymbol{x}_1) \geq g(\boldsymbol{x}_2) + \langle \nabla g(\boldsymbol{x}_2), \boldsymbol{x}_1 - \boldsymbol{x}_2 \rangle + \frac{\rho}{2}\|\boldsymbol{x}_1 - \boldsymbol{x}_2\|_2^2$. Moreover, we say $g(\boldsymbol{x})$ is $\ell$-smooth if there exists a positive constant $\ell$ such that $\|\nabla g(\boldsymbol{x}_1) - \nabla g(\boldsymbol{x}_2)\|_2 \leq \ell\|\boldsymbol{x}_1 - \boldsymbol{x}_2\|_2$.*

In all our analysis, we will impose the basic Assumption 1 to bound stochastic gradient variance.

**Assumption 1** (Bounded gradient). *For each loss $f_i(\boldsymbol{x})$, the distance between its gradient $\nabla f_i(\boldsymbol{x})$ and the full gradient $\nabla f(\boldsymbol{x})$ is upper bounded as $\max_i \|\nabla f_i(\boldsymbol{x}) - \nabla f(\boldsymbol{x})\|_2 \leq G$.*

If $f_i(\boldsymbol{x})$ is $\ell$-smooth and the domain of interest $\boldsymbol{\mathcal{X}}$ is bounded, then the bounded gradient assumption can be naturally implied. We explicitly write out this assumption for the sake of notation simplicity. Following [5, 20, 21], we also employ the incremental first order oracle (IFO) complexity as the computational complexity metric for solving the finite-sum minimization problem (1).

**Definition 2.** *An IFO takes an index $i \in [n]$ and a point $\boldsymbol{x} \in \boldsymbol{\mathcal{X}}$, and returns the pair $(f_i(\boldsymbol{x}), \nabla f_i(\boldsymbol{x}))$.*

The IFO complexity can more accurately reflect the overall computational performance of a first-order algorithm, as objective value and gradient evaluation usually dominate the per-iteration complexity.

**Algorithm 1** Hybrid SGD under WoRS

---

**Input:** Initial point $\boldsymbol{x}^0$, sample index set $\mathcal{S} = \{1, \cdots, n\}$, learning rate $\{\eta_k\}$, mini-batch size $\{s_k\}$.
**for** $k = 0$ **to** $T - 1$ **do**
    Select $s_k$ samples $\mathcal{S}_k$ by WoRS from $\mathcal{S} - \bigcup_{i=0}^{k-1} \mathcal{S}_i$.
    Compute the gradient $\boldsymbol{g}^k = \frac{1}{s_k} \sum_{i_k \in \mathcal{S}_k} \nabla f_{i_k}(\boldsymbol{x}^k)$.
    Update $\boldsymbol{x}^{k+1} = \Phi_{\boldsymbol{\mathcal{X}}}\left(\boldsymbol{x}^k - \eta_k \boldsymbol{g}^k\right)$.
**end for**
**Output:** $\boldsymbol{x}^a$ sampled uniformly from $\{\boldsymbol{x}^k\}_{k=0}^{T-1}$ for strong convex and linearly structured problems or $\{\boldsymbol{x}^k\}_{k=\lfloor 0.5T \rfloor}^{T-1}$ for non-strongly/non-convex problems.

---

## 3 General Analysis for HSGD under WoRS

The WoRS-based HSGD algorithm is outlined in Algorithm 1. Here we systematically analyze its convergence performance for strongly/non-strongly convex and non-convex problems. Similar to [13], we focus our analysis on the scenario where a single pass (or less) over data is of interest, which occurs, *e.g.* in streaming data analysis. According to our empirical study (see, *e.g.*, Figure 3), running Algorithm 1 for a single pass over data can provide satisfactory accuracy in many cases.

### 3.1 A key lemma

It is well understood that unbiased gradient estimation with gradually vanishing variance is important for accelerating randomized algorithms [5, 15]. This is because the increasingly more accurate estimate of full gradient allows the algorithm to move ahead with more aggressive step-size to decrease the objective value. However, for WoRS implementation, the mini-batch terms selected at each iteration are no longer statistically independent, leading to biased gradient estimate $\boldsymbol{g}^k$, *i.e.*

$$\mathbb{E}[\boldsymbol{g}^k] = \mathbb{E}\Big[\frac{1}{s_k} \sum\nolimits_{i_k \in \mathcal{S}_k} \nabla f_{i_k}(\boldsymbol{x}^k)\Big] \neq \nabla f(\boldsymbol{x}^k).$$

Such a biased estimate $\boldsymbol{g}^k$ brings a challenge to bounding its variance $\mathbb{E}\|\boldsymbol{g}^k - \nabla f(\boldsymbol{x}^k)\|_2^2$ with common techniques such as Bernstein inequality [22] and those existing bounds on $\mathbb{E}\|\boldsymbol{g}^k - \nabla f(\boldsymbol{x}^k)\|_2^2$ under WRS [23]. To tackle this challenge, we introduce the following sequence of random variables:

$$\boldsymbol{z}_k = \bar{\boldsymbol{\mu}}_k - \nabla f(\boldsymbol{x}^k) \quad \text{and} \quad \boldsymbol{z}_0 = \boldsymbol{0},$$

where $\bar{\boldsymbol{\mu}}_k := \frac{1}{s_k'} \sum_{i_k \in \mathcal{S}_k'} \nabla f_{i_k}(\boldsymbol{x}^k)$, $\mathcal{S}_k' = \mathcal{S} - \bigcup_{i=0}^{k-1} \mathcal{S}_i$ and $s_k' = n - \sum_{i=0}^{k-1} s_i$ in which $\mathcal{S} = \{1, 2, \cdots, n\}$ denotes the index set of all samples. We can prove $\boldsymbol{z}_k$'s form a martingale, *i.e.* $\mathbb{E}[\boldsymbol{z}_k \mid \boldsymbol{z}_{k-1}, \ldots, \boldsymbol{z}_0] = \boldsymbol{z}_{k-1}$. Moreover, we can show that its squared Euclidian norm is bounded by

$$\mathbb{E}\left[\|\boldsymbol{z}_k\|_2^2 \mid \boldsymbol{z}_{k-1}, \ldots, \boldsymbol{z}_0\right] \leq \frac{4G^2}{n - b_k}\Big[1 - \frac{(n - b_k)^2 - b_k}{n(n - b_k)}\Big],$$

where $b_k = \sum_{i=0}^{k-1} s_i$. Similarly, we define a sequence of $\bar{\boldsymbol{z}}_i$ for the process of without-replacement sampling a subset $\widehat{\mathcal{S}}_i$ of size $\widehat{s}_i$ from $\mathcal{S}_k'$ of size $s_k'$:

$$\bar{\boldsymbol{z}}_i = \frac{1}{\widehat{s}_i} \sum_{i_k \in \widehat{\mathcal{S}}_i} \nabla f_{i_k}(\boldsymbol{x}^k) - \bar{\boldsymbol{\mu}}_k \quad \text{and} \quad \bar{\boldsymbol{z}}_0 = \boldsymbol{0}.$$

Also, we can prove that $\bar{\boldsymbol{z}}_i$ is a martingale with bounded norm:

$$\mathbb{E}\left[\|\bar{\boldsymbol{z}}_i\|_2^2 \mid \bar{\boldsymbol{z}}_{i-1}, \ldots, \bar{\boldsymbol{z}}_0\right] \leq \frac{4G^2}{\widehat{s}_i}\Big[1 - \frac{\widehat{s}_i - 1}{s_k'}\Big].$$

Based on the above arguments, we formulate the $k$-th WoRS as a stochastic process consisting of two phases. In the first phase, we are given $s_k'$ samples indexed by $\mathcal{S}_k' = \mathcal{S} - \bigcup_{i=0}^{k-1} \mathcal{S}_i$ after $k - 1$ times of WoRS over all the data. The sampling result is recorded by $\boldsymbol{z}_k$. Then, in the second phase, we sample $s_k$ data from the remaining $s_k'$ samples indexed by $\mathcal{S}_k'$ in a without-replacement fashion,

which corresponds to $\bar{z}_i$. Based on such a WoRS process, we have

$$\mathbb{E}[\|g^k - \nabla f(x^k)\|_2^2] \leq 2\mathbb{E}[\|\bar{\mu}_k - \nabla f(x^k)\|_2^2 + \|g^k - \bar{\mu}_k\|_2^2]$$

$$= 2\mathbb{E}[\|z_k\|_2^2 \mid z_{k-1}, \ldots, z_0] + 2\mathbb{E}[\|\bar{z}_{s_k}\|_2^2 \mid \bar{z}_{s_k-1}, \ldots, \bar{z}_0; z_{k-1}, \ldots, z_0]$$

$$\leq \frac{8G^2}{n - b_k}\left[1 - \frac{(n - b_k)^2 - b_k}{n(n - b_k)}\right] + \frac{8G^2}{s_k}\left[1 - \frac{s_k - 1}{n - b_k}\right].$$

The above claim leads to the following Lemma 1 which is key to our WoRS-based convergence analysis in the sections to follow. Notice, the above results on gradient variance are some intermediate results for proving Lemma 1, whose full proofs are deferred to Appendix A.

**Lemma 1.** *The gradient $g^k$ estimated by WoRS in Algorithm 1 satisfies $\mathbb{E}\left[\|g^k - \nabla f(x^k)\|_2^2\right] \leq \frac{24G^2}{s_k}$.*

From Lemma 1, we find that the gradient variance $\mathbb{E}[\|g^k - \nabla f(x^k)\|_2^2]$ in Algorithm 1 is controlled by $1/s_k$. Accordingly, the estimated gradient becomes increasingly more accurate and stable. This means that by gradually increasing the mini-batch size, HSGD under WoRS can reduce variance, similar to SVRG and SAGA, but without requiring to integrate historical gradients or full gradient of the snapshot point into current gradient estimate. In the following sections, we will extensively use Lemma 1 to analyze HSGD under WoRS.

By applying Bernstein inequality, Friedlander *et al.* [6] showed that $\mathbb{E}[\|g^k - \nabla f(x^k)\|_2^2] = \mathcal{O}\left(\frac{n - s_k}{n s_k}\right)$ if the $s_k$ samples selected at iteration $k$ are different, but are sampled from the *entire* data set. In contrast, our considered WoRS strategy assumes the $s_k$ different samples are drawn from the *remaining* set $\mathcal{S} - \cup_{i=0}^{k-1}\mathcal{S}_i$, and thus needs to take into account the statistical dependence among iterations to bound the stochastic gradient variance.

## 3.2 Strongly convex functions

We analyze the convergence behavior of both the computed solution $x$ and the objective $f(x)$ under the strongly convex setting. Our convergence result on the computed solution is stated in Theorem 1.

**Theorem 1.** *Suppose $f(x)$ is $\rho$-strongly-convex and each $f_i(x)$ is $\ell$-smooth. With learning rate $\eta_k = \frac{\rho}{\ell^2}$ and mini-batch size $s_k = \frac{\tau}{\zeta^k}$ where $\zeta = 1 - \frac{\rho}{18\ell^2}$ and $\tau \geq \frac{G^2}{\|x^0 - x^*\|^2}\max\left(\frac{324}{\rho^2}, \frac{432}{\ell^2}\right)$, we have*

$$\mathbb{E}\|x^a - x^*\|_2^2 \leq \left(1 - \frac{\rho^2}{18\ell^2}\right)^T\|x^0 - x^*\|_2^2,$$

*where $x^a$ is the output solution of Algorithm 1 and $T$ is the number of iterations.*

A proof of this result is given in Appendix B.1. From Theorem 1, if mini-batch size is increased at an exponential rate $\frac{1}{1-\gamma}$ with $\gamma = \frac{\rho}{18\ell^2}$, then the objective in HSGD converges linearly at the rate of $\mathcal{O}\left((1 - \gamma)^k\right)$ for strongly convex problems. This implies that HSGD enjoys the merits of both SGD and FGD. Specifically, similar to SGD, the per-iteration computation of HSGD is cheap as it is free of computing the full gradient $\nabla f(x)$. Meanwhile, it uses a constant learning rate and enjoys the steady convergence rate of FGD. As the condition number $\kappa = \ell/\rho$ is usually large in realistic problems, the exponential rate $\frac{1}{1-\gamma}$ is actually only slightly above one. This means even a moderate-scale dataset allows plenty of HSGD iterations in one epoch to decrease the objective value sufficiently, as illustrated in Figure 2 and 3. Friedlander *et al.* [6] proved that HSGD has linear convergence rate under WRS. Theorem 1 generalizes the result to WoRS. Then we can derive the IFO complexity of HSGD for strongly-convex problems in the following corollary, for which proof is given in Appendix B.2.

**Corollary 1.** *Suppose the assumptions in Theorem 1 hold. To achieve $\mathbb{E}\|x^a - x^*\|_2^2 \leq \epsilon$, the IFO complexity of HSGD is $\mathcal{O}\left(\frac{\kappa^2 G^2}{\epsilon}\right)$ where $\kappa = \frac{\ell}{\rho}$ denotes the condition number of the objective $f(x)$.*

From Corollary 1, the IFO complexity of HSGD for strongly convex problems is at the order of $\mathcal{O}\left(\frac{\kappa^2}{\epsilon}\right)$, which is not relying on the sample size $n$. So when $n$ dominates $\frac{1}{\epsilon}$, HSGD can be superior to the algorithms with complexity linearly relying on $n$, such as SVRG and SAGA.

Gürbüzbalaban *et al.* [9] showed that by processing each individual $f_i(x)$ with random shuffling at each iteration and adopting a diminishing learning rate $\eta_k = \mathcal{O}\left(\frac{1}{k^\beta}\right)$ with $\beta \in (\frac{1}{2}, 1)$, the IFO

complexity of FGD is $\mathcal{O}\left(\kappa^2 \frac{n}{\epsilon}\right)$ for achieving $\mathbb{E}\|\boldsymbol{x}^a - \boldsymbol{x}^*\|_2^2 \le \epsilon$. So HSGD is $n$ times faster than FGD. This is because at each iteration, unlike FGD requiring to access all data, HSGD only samples a mini-batch for gradient estimation without sacrificing convergence rate. Ying *et al.* [12] proved that under WoRS, both SAGA and AVRG converge linearly and have IFO complexity of $\mathcal{O}\left(n\kappa^2 \log\left(\frac{1}{\epsilon}\right)\right)$. Hence, HSGD will outperform SAGA and AVRG if $n$ dominates $\frac{1}{\epsilon}$, which is usually the case when the data scale is huge while the desired accuracy $\epsilon$ is moderately small (*e.g.* $1/\sqrt{n}$).

Shamir [13] proved that for linearly structured problems, SGD under WoRS has IFO complexity $\mathcal{O}\left(\frac{\kappa}{\epsilon} \log\left(\frac{\kappa}{\epsilon}\right)\right)$ by measuring the objective (see Section 4). Here we can also establish the shaper convergence behavior of the objective value. The result is presented in Theorem 2 with proof provided in Appendix B.3.

**Theorem 2.** *Assume $f(\boldsymbol{x})$ is $\rho$-strongly-convex and each $f_i(\boldsymbol{x})$ is $\ell$-smooth. Let learning rate $\eta_k = \frac{1}{\ell}$ and mini-batch size $s_k = \frac{\tau}{\zeta^k}$ with $\zeta = 1 - \frac{\rho}{2\ell}$ and $\tau \ge \frac{6G^2}{\rho[f(\boldsymbol{x}^0) - f(\boldsymbol{x}^*)]}$. Then the output $\boldsymbol{x}^a$ of Algorithm 1 satisfies*

$$\mathbb{E}\left[f(\boldsymbol{x}^a) - f(\boldsymbol{x}^*)\right] \le \left(1 - \frac{\rho}{2\ell}\right)^T (f(\boldsymbol{x}^0) - f(\boldsymbol{x}^*)).$$

*Moreover, to achieve $\mathbb{E}[f(\boldsymbol{x}^a) - f(\boldsymbol{x}^*)] \le \epsilon$, the IFO complexity of HSGD is $\mathcal{O}\left(\frac{\kappa G^2}{\epsilon}\right)$, where $\kappa = \frac{\ell}{\rho}$.*

Theorem 2 shows that HSGD also enjoys linear convergence rate on the objective by using exponentially mini-batch size. But it has lower complexity $\mathcal{O}\left(\frac{\kappa}{\epsilon}\right)$ under the measurement $\mathbb{E}[f(\boldsymbol{x}^a) - f(\boldsymbol{x}^*)] \le \epsilon$ which is in contrast to the complexity $\mathcal{O}\left(\frac{\kappa^2}{\epsilon}\right)$ for achieving $\mathbb{E}\|\boldsymbol{x}^a - \boldsymbol{x}^*\|_2^2 \le \epsilon$. This is because the objective analysis allows to use more aggressive step-size $\frac{1}{\ell}$, while the analysis on the solution requires smaller learning rate $\frac{\rho}{\ell^2}$. In this way, HSGD with larger step-size converges faster.

### 3.3 Non-strongly convex functions

We proceed to analyze the convergence performance of HSGD for non-strongly convex problems. Our result for this case is summarized in Theorem 3. To our best knowledge, this is the first convergence guarantee of WoRS-based methods for *non-strongly* convex problems.

**Theorem 3.** *Suppose $f(\boldsymbol{x})$ is convex and each $f_i(\boldsymbol{x})$ is $\ell$-smooth. Assume that $\|\boldsymbol{x}_1 - \boldsymbol{x}_2\|_2 \le D$ holds for $\forall \boldsymbol{x}_1, \boldsymbol{x}_2 \in \boldsymbol{\mathcal{X}}$. Then with the learning rate $\eta_k = \frac{1}{2\ell}$ and mini-batch size $s_k = (k+1)^2$, we have*

$$\mathbb{E}[f(\boldsymbol{x}^a) - f(\boldsymbol{x}^*)] \le \frac{4\ell D^2 + 24GD}{T} + \frac{48G^2}{\ell T^2},$$

*where $\boldsymbol{x}^a$ denotes the output solution of Algorithm 1 and $T$ is the number of iterations.*

A proof of this result is given in Appendix B.4. Theorem 3 shows that if one expands the mini-batch size at $\mathcal{O}\left(k^2\right)$, then the convergence rate of HSGD under WoRS is $\mathcal{O}\left(\frac{1}{T}\right)$. In [13], a sub-linear rate was established for WoRS-based SGD in a special class of convex problems with $f_i(\boldsymbol{x}) = h_i(\langle \boldsymbol{a}_i, \boldsymbol{x} \rangle)$. A detailed comparison between their result and ours for such a structured formulation will be discussed in Section 4. Under the assumption $\sum_{k=0}^{+\infty} \|\boldsymbol{g}^k - \nabla f(\boldsymbol{x}^k)\|_2 < +\infty$, Friedlander *et al.* [6] showed that WRS-based HSGD outputs $f(\boldsymbol{x}^a) - f(\boldsymbol{x}^*) = \mathcal{O}\left(\frac{1}{T}\right)$. However, such an assumption holds only if HSGD selects at least $\mathcal{O}\left(k^2\right)$ samples at the $k$-th iteration due to $\mathbb{E}[\|\boldsymbol{g}^k - \nabla f(\boldsymbol{x}^k)\|_2^2] = \mathcal{O}\left(\frac{n - s_k}{n s_k}\right)$. In this way, their result under WRS is of the same order as ours under WoRS. The following corollary gives the corresponding IFO complexity. A proof of this result is given in Appendix B.5.

**Corollary 2.** *Suppose the assumptions in Theorem 3 hold. To achieve $\mathbb{E}[f(\boldsymbol{x}^a) - f(\boldsymbol{x}^*)] \le \epsilon$, the IFO complexity of HSGD is $\mathcal{O}\left(\frac{(6GD + \ell D^2)^3}{\epsilon^3}\right)$.*

### 3.4 Non-convex functions

Now we analyze HSGD for non-convex problems, which to our knowledge has not yet been addressed elsewhere in literature. The result is stated in Theorem 4 with proof provided in Appendix B.6.

**Theorem 4.** *Suppose each $f_i(\boldsymbol{x})$ is $\ell$-smooth and for $\forall \boldsymbol{x}_1, \boldsymbol{x}_2 \in \boldsymbol{\mathcal{X}}$, $\|\boldsymbol{x}_1 - \boldsymbol{x}_2\|_2 \le D$. With learning rate $\eta_k = \frac{1}{2\ell}$ and mini-batch size $s_k = k+1$, the output $\boldsymbol{x}^a$ of Algorithm 1 with $T$ iterations satisfies*

$$\mathbb{E}\left[\|\nabla f(\boldsymbol{x}^a)\|_2^2\right] \le \frac{4\ell^2 D^2 + 35G^2}{T}.$$

Theorem 4 guarantees that for non-convex problems, HSGD exhibits $\mathcal{O}\left(\frac{1}{T}\right)$ rate of convergence by linearly expanding the mini-batch size at each iteration. Here we follow the convention in [10, 11, 23] to adopt the value $\|\nabla f(\boldsymbol{x}^a)\|_2^2$ as a measurement of quality for approximate stationary solutions. Then we drive the IFO complexity of HSGD in the following corollary with proof in Appendix B.7.

**Corollary 3.** *Suppose the assumptions in Theorem 4 hold. To achieve* $\mathbb{E}\left[\|\nabla f(\boldsymbol{x}^a)\|_2^2\right] \leq \epsilon$, *the IFO complexity of the HSGD in Algorithm 1 is* $\mathcal{O}\left(\frac{(4\ell^2 D^2 + 35G^2)^2}{\epsilon^2}\right)$.

The IFO complexity for non-convex problems looks lower than that for non-strongly convex ones in Corollary 2. This is because we use $\mathbb{E}\left[f(\boldsymbol{x}^a - f(\boldsymbol{x}^*)\right] \leq \epsilon$ as sub-optimality measurement for arbitrary convex problems and $\mathbb{E}\left[\|\nabla f(\boldsymbol{x}^a)\|_2^2\right] \leq \epsilon$ for non-convex problems.

# 4 Analysis for Linearly Structured Problems

We further consider a special case of problem (1) where each $f_i(\boldsymbol{x})$ has a linear prediction structure:

$$f(\boldsymbol{x}) := \frac{1}{n} \sum\nolimits_{i=1}^n f_i(\boldsymbol{x}), \text{ where } f_i(\boldsymbol{x}) = h(\langle \boldsymbol{a}_i, \boldsymbol{x} \rangle). \tag{2}$$

Here $\boldsymbol{a}_i$ denotes the $i$-th sample vector and $h(\cdot)$ denotes a convex loss function. Such a formulation covers several common problems in machine learning, such as $f_i(\boldsymbol{x}) = \frac{1}{2}(b_i - \boldsymbol{a}_i^\top \boldsymbol{x})^2$ for least square regression and $f_i(\boldsymbol{x}) = \log(1 + \exp(-b_i \boldsymbol{a}_i^\top \boldsymbol{x}))$ for logistic regression, where $b_i$ is the real or binary target output. Such a special problem setting has been considered in [13] for analyzing SGD under WoRS. To make a comprehensive comparison, we specify our strongly convex analysis to (2), and improve our non-strongly-convex results when the surrogate loss $h(\cdot)$ is strongly convex.

**Strongly convex case.** In this case, according to Theorem 2, HSGD converges linearly and its IFO complexity is $\mathcal{O}\left(\frac{\kappa}{\epsilon}\right)$. By comparison, SGD under WoRS in [13] converges at $\mathcal{O}\left(\frac{\kappa}{T}\log\left(\frac{1}{T}\right)\right)$ and has IFO complexity $\mathcal{O}\left(\frac{\kappa}{\epsilon}\log\left(\frac{\kappa}{\epsilon}\right)\right)$, slightly higher than ours due to the presence of the factor $\log\left(\frac{\kappa}{\epsilon}\right)$. Moreover, it is allowed in HSGD to use constant step-size which is required to be shrinking in [13].

On this special problem, other results on general strongly convex problems can also be applied. As discussed in Section 3.2, HSGD is $n$ times faster than FGD [9], and is superior to SAGA [12] and AVRG [12] when $n$ dominates $\frac{1}{\epsilon}$. Shamir [13] showed that SVRG [5] under WoRS has IFO complexity $\mathcal{O}\left(\left(n + \kappa \log\left(\frac{1}{\epsilon}\right)\right) \log\left(\frac{1}{\epsilon}\right)\right)$ in ridge regression with the measurement $\mathbb{E}[f(\boldsymbol{x}) - f(\boldsymbol{x}^*)]$. Comparatively, such an IFO complexity is still higher than HSGD when sample size $n$ is large and the desired accuracy is moderately small.

**Non-strongly convex case with strongly-convex $h(\cdot)$.** When the loss $f(\boldsymbol{x})$ in (2) is non-strongly convex but the surrogate loss $h(\cdot)$ is strongly convex, we show an improved convergence rate in Theorem 5 than that in Theorem 3 for general cases. See proof of Theorem 5 in Appendix C.1.

**Theorem 5.** *Suppose* $f_i(\boldsymbol{x}) = h(\boldsymbol{a}_i^\top \boldsymbol{x})$ *is* $\ell$-*smooth and* $h(\cdot)$ *is* $\alpha$-*strongly convex. Let* $\sigma(\boldsymbol{A})$ *denote the smallest non-zero singular value of the matrix* $\boldsymbol{A} = [\boldsymbol{a}_1^\top; \boldsymbol{a}_2^\top; \ldots, \boldsymbol{a}_n^\top]$ *and* $\mu = \alpha\sigma(\boldsymbol{A})$. *If the learning rate* $\eta_k = \frac{1}{\ell}$ *and mini-batch size* $s_k = \frac{\tau}{\zeta^k}$ *with* $\tau \geq \frac{24G^2}{\mu[f(\boldsymbol{x}^0) - f(\boldsymbol{x}^*)]}$ *and* $\zeta = 1 - \frac{\mu}{2\ell}$, *we have*

$$\mathbb{E}\left[f(\boldsymbol{x}^a) - f(\boldsymbol{x}^*)\right] \leq \left(1 - \frac{\mu}{2\ell}\right)^T (f(\boldsymbol{x}^0) - f(\boldsymbol{x}^*)),$$

*where* $\boldsymbol{x}^a$ *denotes the output solution of Algorithm 1 and* $T$ *is the number of iterations.*

Theorem 5 shows if the function $h(\boldsymbol{a}_i^\top \boldsymbol{x})$ is strongly convex in terms of the linear prediction $\boldsymbol{a}_i^\top \boldsymbol{x}$, by exponentially sampling the data at each iteration, HSGD converges linearly even though $f(\boldsymbol{x})$ might be non-strongly convex. Based on Theorem 5, we further derive the IFO complexity of Algorithm 1 for such a special problem, as summarized in Corollary 4 with proof in Appendix C.2.

**Corollary 4.** *Suppose the assumptions in Theorem 5 hold. To achieve* $\mathbb{E}[f(\boldsymbol{x}^a) - f(\boldsymbol{x}^*)] \leq \epsilon$ *for the special problem, the IFO complexity of the proposed algorithm is* $\mathcal{O}\left(\frac{\ell G^2}{\mu^2 \epsilon}\right)$.

It is interesting to compare Theorem 5 and Corollary 4 with those existing ones for SGD. Particularly, it was shown by Shamir [13] that $\mathbb{E}\left[f(\boldsymbol{x}^a) - f(\boldsymbol{x}^*)\right] \leq R_T/T + 2(12 + \sqrt{2}D)/\sqrt{n}$ for SGD, where $R_T$ is the regret bound of SGD on problem (2), at the order of $\mathcal{O}(D\ell\sqrt{T})$. This gives a

convergence rate of $\mathcal{O}(1/\sqrt{T})$ and IFO complexity of $\mathcal{O}(1/\epsilon^2)$. However, there exists an accuracy barrier $\mathcal{O}(1/\sqrt{n})$ due to the statistical error term $2(12+\sqrt{2}D)/\sqrt{n}$ which is the artifact brought by analyzing the regret. In sharp contrast, our result in Theorem 5 guarantees that HSGD converges to the global optimum of problem (2). More importantly, provided that $h(\cdot)$ is strongly convex, HSGD has superior IFO complexity of $\mathcal{O}\left(\frac{1}{\epsilon}\right)$ to the SGD complexity $\mathcal{O}\left(\frac{1}{\epsilon^2}\right)$ given in [13].

## 5 Experiments

We compare HSGD with several state-of-the-art algorithms, including SGD [2], SVRG [5], SAGA [15], AVRG [12] and SCGC [23], under WoRS for all. We consider two sets of learning tasks. The first contains two convex problems: $\ell_2$-regularized logistic regression $\min_{\boldsymbol{x}} \frac{1}{n} \sum_{i=1}^{n} \left[ \log(1+\exp(-b_i \boldsymbol{a}_i^\top \boldsymbol{x})) + \frac{\lambda}{2} \|\boldsymbol{x}\|_2^2 \right]$ and $k$-classes softmax regression $\min_{\boldsymbol{x}} \frac{1}{n} \sum_{i=1}^{n} \sum_{j=1}^{k} \left[ \frac{\lambda}{2k} \|\boldsymbol{x}_j\|_2^2 - \mathbf{1}\{b_i = j\} \log \frac{\exp(\boldsymbol{a}_i^\top \boldsymbol{x}_j)}{\sum_{l=1}^{k} \exp(\boldsymbol{a}_i^\top \boldsymbol{x}_l)} \right]$, where $b_i$ is the target output of $\boldsymbol{a}_i$. The other one is a non-convex problem of training multi-layer neural networks. We run simulations on 10 datasets (see Appendix D). Hyper-parameters of all the algorithms are tuned to best.

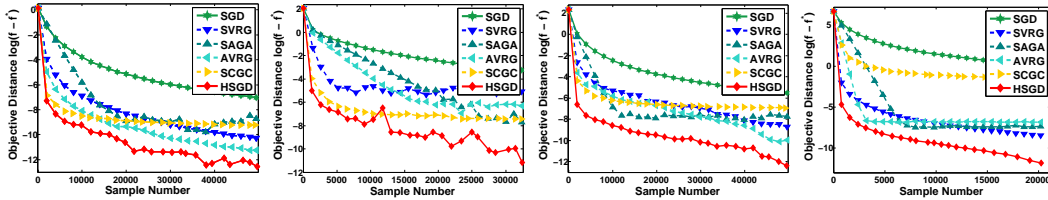

(a) Logistic regression. From left to right: ijcnn, A09, w08 and rcv11.

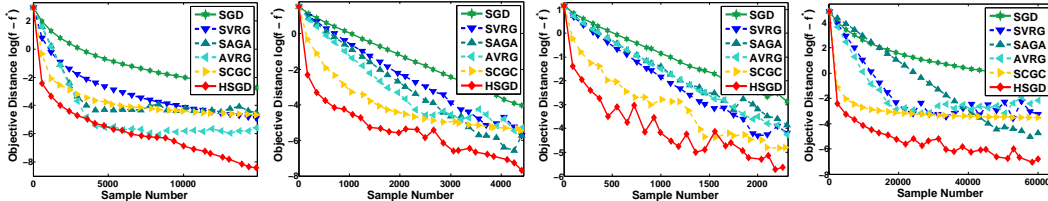

(b) Softmax regression. From left to right: protein, satimage, sensorless and mnist.

Figure 2: Single-epoch processing: comparison of randomized algorithms for a single pass over data.

### 5.1 Convex problems

As the first set of problems are strongly convex, we follow Theorem 2 to exponentially expand the mini-batch size $s_k$ in HSGD with $\tau = 1$. We run FGD until the gradient $\|\nabla f(\boldsymbol{x})\|_2 \le 10^{-10}$. Then use the output as the optimal value $f^*$ for sub-optimality estimation in Figure 1 (a), 2 and 3.

**Single-epoch processing in well-conditioned problems.** We first consider the case where the optimization problem is well-conditioned with strong regularization, such that good results can be obtained after only one epoch of data pass. Single-epoch learning is common in online learning. For two problems, we respectively set their regularization parameters to $\lambda = 0.01$ and $\lambda = 0.1$.

Figure 2 summarizes the numerical results. On the simulated well-conditioned tasks most algorithms achieve high accuracy after one epoch, while HSGD (WoRS) converges much faster. This confirms Corollary 1 that HSGD is cheaper in IFO complexity $(\mathcal{O}\left(\frac{\kappa^2}{\epsilon}\right))$ than other considered variance-reduced algorithms $(\mathcal{O}\left(n\kappa^2 \log\left(\frac{1}{\epsilon}\right)\right))$ when the desired accuracy is moderately low and data size is large.

**Multi-epoch processing in ill-conditioned problems.** To solve more challenging problems, a method usually needs multiple cycles of data processing to reach high accuracy solution. Thus we develop a practical implementation of HSGD for multiple epochs processing. After visiting all data in one full pass, it continues to increase the mini-batch size, allowing possible with-replacement sampling, until $s_k > n$. After that, HSGD degenerates to standard FGD. But this does not happen in

our testing cases, since we set the exponential rate sufficiently small. To generate more challenging optimization tasks, we reset the regularization strength parameter in softmax regression as $\lambda = 0.001$.

Figure 3 shows that HSGD under WoRS outperforms all compared algorithms. These observations align well with those in Figure 2, implying HSGD has sharper convergence behavior when the sample size $n$ is large and the desired accuracy is moderate. The convergence curves of HSGD also confirm the effectiveness of our practical implementation in continuously decreasing the objective value.

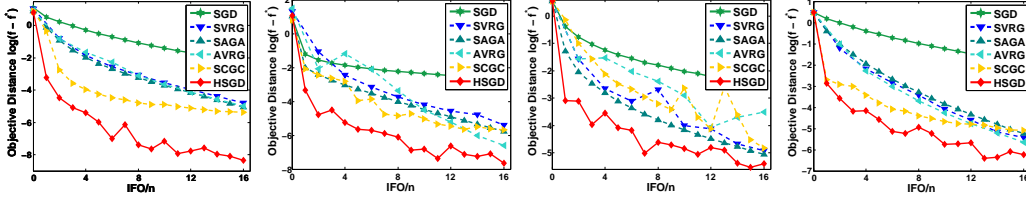

Figure 3: Multi-epoch processing: comparison of randomized algorithms for multiple passes over data (Softmax regression. From left to right: protein, satimage, sensorless and letter).

## 5.2 Non-convex problems

Here we evaluate HSGD for optimizing a three-layer feedforward neural network with a logistic loss on ijcnn1 and covtype and softmax loss on sensorless (see Figure 1 (b)). For both cases we set $\lambda = 0.01$. The network has an architecture of $d - 30 - c$, where $d$ and $c$ respectively denote the input and output dimension and 30 is the neuron number in the hidden layer. We test two versions of HSGD, namely HSGD-lin and HSGD-exp, respectively with linearly and exponentially increasing mini-batch size from $s_0 = 1$. We use the same initialization for all algorithms.

From Figure 4, HSGD-exp exhibits similar convergence behavior as above: it decreases the loss very quickly. Comparatively, HSGD-lin outputs more accurate solutions with linearly increasing batch size, which is consistent with Theorem 4. We note HSGD-lin behaves differently in Figure 4 (a) and (b). In Figure 4 (a), it converges relatively slowly at the beginning, while in Figure 4 (b) much faster, of which we attribute the reason to the different characteristics of data.

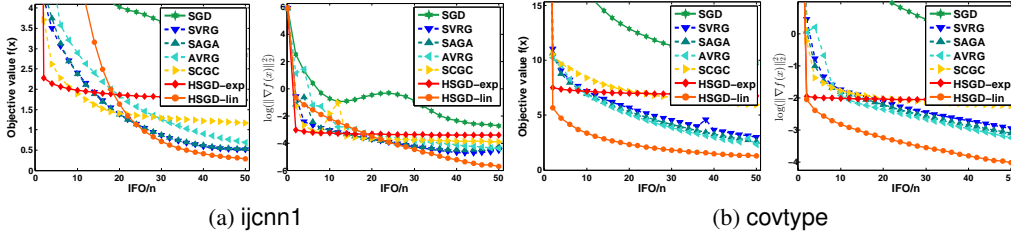

(a) ijcnn1        (b) covtype

Figure 4: Non-convex results: comparison of randomized algorithms on forward neural networks.

## 6 Conclusion

We analyzed the rate-of-convergence of HSGD under WoRS for strongly/arbitrarily convex and non-convex problems. We proved that under WoRS, HSGD with constant step-size can match FG descent in convergence rate, while maintaining comparable sample-size-independent IFO complexity to SGD. Compared to the variance-reduced SGD methods such as SVRG and SAGA, HSGD tends to gain better efficiency and scalability in the setting where the sample size is large while the required optimization accuracy is moderately small. Numerical results confirmed our theoretical results.

## Acknowledgements

Jiashi Feng was partially supported by NUS startup R-263-000-C08-133, MOE Tier-I R-263-000-C21-112, NUS IDS R-263-000-C67-646, ECRA R-263-000-C87-133 and MOE Tier-II R-263-000-D17-112. Xiao-Tong Yuan was supported in part by Natural Science Foundation of China (NSFC) under Grant 61522308 and Grant 61876090, and in part by Tencent AI Lab Rhino-Bird Joint Research Program No.JR201801.

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
