[Supplementary Material]

# New Insight into Hybrid Stochastic Gradient Descent: Beyond With-Replacement Sampling and Convexity (Supplementary File)

**Pan Zhou**\*     **Xiao-Tong Yuan**†     **Jiashi Feng**\*

\* Learning & Vision Lab, National University of Singapore, Singapore
† B-DAT Lab, Nanjing University of Information Science & Technology, Nanjing, China
pzhou@u.nus.edu     xtyuan@nuist.edu.cn     elefjia@nus.edu.sg

## Abstract

This supplementary document contains the technical proofs of convergence results and some additional numerical results of the NeurIPS'18 paper entitled "New Insight into Hybrid Stochastic Gradient Descent: Beyond With-Replacement Sampling and Convexity". It is structured as follows. The proof of our key technical lemma, Lemma 1, is presented in Appendix A, followed by the proofs of main results in Appendices B and C for Section 3 and Section 4, respectively. Some detailed descriptions of data and algorithm along with more numerical results are provided in Appendix D.

## A  Proof of Lemma 1

Before proving Lemma 1 in the manuscript, we first give a useful lemma as stated in Lemma 2.

**Lemma 2.** *Assume that $\boldsymbol{a}_1, \ldots, \boldsymbol{a}_n$ denote the feature vectors of the $n$ samples and let $\{\sigma_{(1)}, \ldots, \sigma_{(n)}\}$ be a permutation over $\{1, \ldots, n\}$ chosen uniformly at random. Let $\bar{\mathcal{S}}_k = \{\sigma_{(1)}, \ldots, \sigma_{(k)}\}$ and $\widetilde{\mathcal{S}}_k = \{\sigma_{(k+1)}, \ldots, \sigma_{(n)}\}$. For brevity, we further define*

$$\widetilde{\boldsymbol{z}}_k = \frac{1}{n-k} \sum_{i_k \in \widetilde{\mathcal{S}}_k} \left( \nabla f_{i_k}(\boldsymbol{x}) - \mu \right) \qquad and \qquad \widetilde{\boldsymbol{z}}_0 = \mathbf{0}$$

$$\bar{\boldsymbol{z}}_k = \frac{1}{k} \sum_{i_k \in \bar{\mathcal{S}}_k} \left( \nabla f_{i_k}(\boldsymbol{x}) - \mu \right) \qquad and \qquad \bar{\boldsymbol{z}}_0 = \mathbf{0},$$

*where $\mu = \nabla f(\boldsymbol{x})$. Then we have*

$$\mathbb{E}\left[\|\widetilde{\boldsymbol{z}}_k\|_2^2\right] \leq \frac{4G^2}{n-k}\left[1 - \frac{(n-k)^2 - k}{n(n-k)}\right], \qquad \mathbb{E}\left[\|\widetilde{\boldsymbol{z}}_k\|_2\right] \leq \frac{2G}{\sqrt{n-k}}\sqrt{1 - \frac{(n-k)^2 - k}{n(n-k)}},$$

$$\mathbb{E}\left[\|\bar{\boldsymbol{z}}_k\|_2^2\right] \leq \frac{4G^2}{k}\left[1 - \frac{k-1}{n}\right], \qquad \mathbb{E}\left[\|\bar{\boldsymbol{z}}_k\|_2\right] \leq \frac{2G}{\sqrt{k}}\sqrt{1 - \frac{k-1}{n}}.$$

*Proof.* Since $\mu = \frac{1}{n}\sum_{i_k=1}^{n} \nabla f_{i_k}(\boldsymbol{x})$, we can establish

$$\widetilde{\boldsymbol{z}}_k = \frac{1}{n-k}\left[-(n-k)\mu + n\mu - \sum_{i_k \in \bar{\mathcal{S}}_k} \nabla f_{i_k}(\boldsymbol{x})\right] = -\frac{1}{n-k} \sum_{i_k \in \bar{\mathcal{S}}_k} \left(\nabla f_{i_k}(\boldsymbol{x}) - \mu\right). \quad (3)$$

On the other hand, we have

$$\mathbb{E}\left[\nabla f_{\sigma_{(k)}}(\boldsymbol{x})|\widetilde{\boldsymbol{z}}_{k-1},\ldots,\widetilde{\boldsymbol{z}}_0\right] = \frac{1}{n-k+1}\sum_{i=k}^{n}\nabla_{\sigma_{(i)}}f(\boldsymbol{x}) = \frac{1}{n-k+1}\left(n\mu - \sum_{i=1}^{k-1}\nabla f_{\sigma_{(i)}}(\boldsymbol{x})\right)$$

$$= \mu - \frac{1}{n-k+1}\sum_{i=1}^{k-1}\left(\nabla f_{\sigma_{(i)}}(\boldsymbol{x}) - \mu\right).$$

$$(4)$$

So we can obtain the following relation between $\mathbb{E}[\widetilde{\boldsymbol{z}}_k]$ and $\widetilde{\boldsymbol{z}}_{k-1}$:

$$\mathbb{E}\left[\widetilde{\boldsymbol{z}}_k|\widetilde{\boldsymbol{z}}_{k-1},\ldots,\widetilde{\boldsymbol{z}}_0\right]$$

$$= -\frac{1}{n-k}\sum_{i=1}^{k-1}\left(\nabla f_{\sigma_{(i)}}(\boldsymbol{x}) - \mu\right) - \frac{1}{n-k}\left(\mathbb{E}\left[\nabla f_{\sigma_{(k)}}(\boldsymbol{x})|\widetilde{\boldsymbol{z}}_{k-1},\ldots,\widetilde{\boldsymbol{z}}_0\right] - \mu\right)$$

$$\overset{①}{=} -\frac{1}{n-k}\sum_{i=1}^{k-1}\left(\nabla f_{\sigma_{(i)}}(\boldsymbol{x}) - \mu\right) - \frac{1}{n-k}\left[\mu - \frac{1}{n-k+1}\sum_{i=1}^{k-1}\left(\nabla f_{\sigma_{(i)}}(\boldsymbol{x}) - \mu\right) - \mu\right]$$

$$= -\frac{1}{n-k+1}\sum_{i=1}^{k-1}\left(\nabla f_{\sigma_{(i)}}(\boldsymbol{x}) - \mu\right)$$

$$\overset{②}{=} \widetilde{\boldsymbol{z}}_{k-1},$$

where ① holds since we plug Eqn. (4) and ② holds due to Eqn. (3). This means that the sequence $\widetilde{\boldsymbol{z}}_k$ is actually a martingale. Meanwhile we have

$$\widetilde{\boldsymbol{z}}_k = \frac{n-k+1}{n-k}\widetilde{\boldsymbol{z}}_{k-1} + \frac{1}{n-k}\left[\nabla f_{\sigma_{(k)}}(\boldsymbol{x}) - \mu\right] = \widetilde{\boldsymbol{z}}_{k-1} + \frac{1}{n-k}\left[\nabla f_{\sigma_{(k)}}(\boldsymbol{x}) - \mu + \widetilde{\boldsymbol{z}}_{k-1}\right].$$

Then we can further bound

$$\mathbb{E}\left[\|\widetilde{\boldsymbol{z}}_k\|^2|\widetilde{\boldsymbol{z}}_{k-1},\ldots,\widetilde{\boldsymbol{z}}_0\right] = \mathbb{E}[\|\widetilde{\boldsymbol{z}}_k - \widetilde{\boldsymbol{z}}_{k-1} + \widetilde{\boldsymbol{z}}_{k-1}\|^2]$$

$$= \mathbb{E}[\|\widetilde{\boldsymbol{z}}_k - \widetilde{\boldsymbol{z}}_{k-1}\|^2 + 2\langle\widetilde{\boldsymbol{z}}_k - \widetilde{\boldsymbol{z}}_{k-1}, \widetilde{\boldsymbol{z}}_{k-1}\rangle + \|\widetilde{\boldsymbol{z}}_{k-1}\|^2]$$

$$= \mathbb{E}\left[\frac{1}{(n-k)^2}\|\nabla f_{\sigma_{(k)}}(\boldsymbol{x}) - \mu + \widetilde{\boldsymbol{z}}_{k-1}\|^2 + \|\widetilde{\boldsymbol{z}}_{k-1}\|^2\right] \qquad (5)$$

$$\overset{①}{\leq} \frac{4G^2}{(n-k)^2} + \|\widetilde{\boldsymbol{z}}_{k-1}\|^2,$$

where ① holds since we have $\|\nabla f_{\sigma_{(k)}}(\boldsymbol{x}) - \mu + \widetilde{\boldsymbol{z}}_{k-1}\|_2 \leq 2(\|\nabla f_{\sigma_{(k)}}(\boldsymbol{x}) - \mu\|^2 + \|\widetilde{\boldsymbol{z}}_{k-1}\|^2) \leq 4G^2$, where $G = \max_i\|\nabla f_i(\boldsymbol{x}) - \mu\|_2$. Conditioned on all the random process and sum Eqn. (5) together, we obtain

$$\mathbb{E}\left[\|\widetilde{\boldsymbol{z}}_k\|^2\right] \leq 4G^2\sum_{i=1}^{k}\frac{1}{(n-i)^2} + \mathbb{E}\|\widetilde{\boldsymbol{z}}_0\|^2 \leq 4G^2\sum_{i=1}^{k}\frac{1}{(n-i)^2} \overset{①}{\leq} \frac{4G^2}{(n-k)^2} + \frac{4(k-1)G^2}{n(n-k)}$$

$$= \frac{4G^2}{n-k}\left(1 - \frac{(n-k)^2 - k}{n(n-k)}\right),$$

where ① holds since for $1 \leq k \leq n$, we have $\sum_{i=k+1}^{n}\frac{1}{i^2} \leq \frac{n-k}{k(n+1)}$. Since the function $\sqrt{\cdot}$ is concave function, we can use Jensen's inequality to obtain

$$\mathbb{E}\left[\|\widetilde{\boldsymbol{z}}_k\|\right] \leq \sqrt{\mathbb{E}\left[\|\widetilde{\boldsymbol{z}}_k\|^2\right]} \leq \frac{2G}{\sqrt{n-k}}\sqrt{1 - \frac{(n-k)^2 - k}{n(n-k)}}.$$

In a similar way, we can prove that $\hat{\boldsymbol{z}}_k = \frac{k}{n-k}\widetilde{\boldsymbol{z}}_k$ is a martingale sequence and

$$\hat{\boldsymbol{z}}_k = \hat{\boldsymbol{z}}_{k-1} + \frac{1}{n-k}\left[\nabla f_{\sigma_{(k)}}(\boldsymbol{x}) - \mu + \hat{\boldsymbol{z}}_{k-1}\right].$$

Therefore, we can bound

$$\mathbb{E}\left[\|\hat{z}_k\|^2\right] \leq \frac{4G^2}{(n-k)^2} + \mathbb{E}\|\hat{z}_{k-1}\|^2 \leq 4G^2 \sum_{i=1}^{k} \frac{1}{(n-i)^2}.$$

So by using $\hat{z}_k = \frac{k}{n-k}\bar{z}_k$, it follows

$$\mathbb{E}\left[\|\bar{z}_k\|^2\right] \leq \frac{4G^2}{k^2} \sum_{i=1}^{k} \frac{(n-k)^2}{(n-i)^2} \leq \frac{4G^2}{k^2}\left(1 + \sum_{i=n-k+1}^{n-1} \frac{(n-k)^2}{i^2}\right) \stackrel{①}{\leq} \frac{4G^2}{k^2}\left(1 + (n-k)^2 \frac{k-1}{n(n-k)}\right)$$

$$\leq \frac{4G^2}{k^2}\left(1 + k - 1 - \frac{k(k-1)}{n}\right) \leq \frac{4G^2}{k}\left(1 - \frac{k-1}{n}\right).$$

Therefore, by Jensen's inequality we have

$$\mathbb{E}\left[\|\bar{z}_k\|\right] \leq \sqrt{\mathbb{E}\left[\|\bar{z}_k\|^2\right]} \leq \frac{2G}{\sqrt{k}}\sqrt{1 - \frac{k-1}{n}}.$$

The proof is completed. $\qquad\square$

Now we use Lemma 2 to prove the following lemma.

**Lemma 3.** *Let $g^k$ be the gradient estimate in Algorithm 1 by WoRS. We have $\mathbb{E}\left[\|g^k - \nabla f(x^k)\|_2^2\right] \leq c_k$, where*

$$c_k = \frac{8G^2}{n - b_k}\left[1 - \frac{(n - b_k)^2 - b_k}{n(n - b_k)}\right] + \frac{8G^2}{s_k}\left[1 - \frac{s_k - 1}{n - b_k}\right],$$

*and $b_k = \sum_{i=0}^{k-1} s_i$.*

*Proof.* Firstly, we introduce the following sequence of random variables $z_k$:

$$z_k = \frac{1}{s_k'} \sum_{i_k \in \mathcal{S}_k'} \left(\nabla f_{i_k}(x^k) - \nabla f(x^k)\right) \quad \text{and} \quad z_0 = 0,$$

where $\mathcal{S}_k' = \mathcal{S} - \bigcup_{i=0}^{k-1} \mathcal{S}_i$ and $s_k' = n - \sum_{i=0}^{k-1} s_i$ respectively denote the indexes and number of remaining samples after the $(k-1)$-th without-replacement sampling in which $\mathcal{S} = \{1, 2, \cdots, n\}$. So actually $z_k$ is actually equivalent to $\widetilde{z}_{b_k}$ where $b_k = \sum_{i=1}^{k-1} s_i$ due to the definition of $\widetilde{z}_{b_k}$ in Lemma 2:

$$\widetilde{z}_{b_k} = \frac{1}{n - b_k} \sum_{i_k \in \widetilde{\mathcal{S}}_k} \left(\nabla f_{i_k}(x) - \nabla f(x)\right) \quad \text{and} \quad \widetilde{z}_0 = 0,$$

where $\widetilde{\mathcal{S}}_k = \mathcal{S} - \bigcup_{i=1}^{k-1} \mathcal{S}_i$. This is because that both $z_k$ and $\widetilde{z}_{b_k}$ actually measure the gradient variance of the data points indexed by $\mathcal{S}_k' = \mathcal{S} - \bigcup_{i=0}^{k-1} \mathcal{S}_i$ which is sampled by WoRS. The only difference is that in the sequence $z_k$, we sample the data $\mathcal{S}_k' = \mathcal{S} - \bigcup_{i=0}^{k-1} \mathcal{S}_i$ by removing mini-batch $\mathcal{S}_k$ at the $k$-th iteration, while in $\widetilde{z}_{b_k}$ in Lemma 2, we sample the data $\widetilde{\mathcal{S}}_k = \mathcal{S} - \bigcup_{i=1}^{k-1} \mathcal{S}_i$ by removing one data in one sampling operation under WoRS. Since both sequences use without-replacement sampling, they have the same gradient variance when the sampled data have the same number. So we can use the bound of $\widetilde{z}_{b_k}$ to bound $z_k$. Thus, by Lemma 2, we can obtain that $\widetilde{z}_{b_k}$ is a martingale (namely, $\mathbb{E}[\widetilde{z}_k | \widetilde{z}_{k-1}, \ldots, \widetilde{z}_0] = \widetilde{z}_{k-1}$) and its norm can be bounded as

$$\mathbb{E}\left[\|z_k\|_2 | z_{k-1}, \ldots, z_0\right] = \mathbb{E}[\|\widetilde{z}_k\|_2 | \widetilde{z}_{k-1}, \ldots, \widetilde{z}_0] \leq \frac{2G}{\sqrt{n - b_k}}\sqrt{1 - \frac{(n - b_k)^2 - b_k}{n(n - b_k)}},$$

$$\mathbb{E}\left[\|z_k\|_2^2 | z_{k-1}, \ldots, z_0\right] = \mathbb{E}[\|\widetilde{z}_k\|_2^2 | \widetilde{z}_{k-1}, \ldots, \widetilde{z}_0] \leq \frac{4G^2}{n - b_k}\left[1 - \frac{(n - b_k)^2 - b_k}{n(n - b_k)}\right]. \tag{6}$$

On the other hand, we define a sequence of $\bar{z}_i$ for the process of without-replacement sampling a subset $\widehat{\mathcal{S}}_i$ of size $\widehat{s}_i$ from $\mathcal{S}'_k$ of size $s'_k$:

$$\bar{z}_i = \frac{1}{\widehat{s}_i} \sum_{i_k \in \widehat{\mathcal{S}}_i} \nabla f_{i_k}(\boldsymbol{x}^k) - \bar{\boldsymbol{\mu}}_k \quad \text{and} \quad \bar{z}_0 = \boldsymbol{0},$$

where $\widehat{s}_i$ actually equals to $s_k$. Then we can use the result in Lemma 2 on $\bar{z}_i$ to bound its norm:

$$\mathbb{E}\left[\|\bar{z}_i\|_2|\bar{z}_{i-1},\ldots,\bar{z}_0\right] \leq \frac{2G}{\sqrt{\widehat{s}_i}}\sqrt{1 - \frac{\widehat{s}_i - 1}{s'_k}} \quad \text{and} \quad \mathbb{E}\left[\|\bar{z}_i\|_2^2|\bar{z}_{i-1},\ldots,\bar{z}_0\right] \leq \frac{4G^2}{\widehat{s}_i}\left[1 - \frac{\widehat{s}_i - 1}{s'_k}\right]. \tag{7}$$

Finally, we combine these two bounds together to obtain our final results. We can formulate the $k$-th without-replacement sampling as a random process, including two phases. In the first phase, we view the remaining samples after the first $k-1$ without-replacement sampling as a without-replacement sampling. In this case, we obtain $s'_k$ samples indexed by $\mathcal{S}'_k = \mathcal{S} - \bigcup_{i=1}^{k-1} \mathcal{S}_i$. This sampling step corresponds to the martingale $z_i$. Then, in the second phase, we sample $s_k$ data from the remaining $s'_k$ samples indexed by $\mathcal{S}'_k$, which corresponds to the martingale sequence $\bar{z}_i$. Define $\bar{\mu} = \frac{1}{s'_k}\sum_{i_k \in \mathcal{S}'_k} \nabla f_{i_k}(\boldsymbol{x}^k)$. Then we can bound

$$\begin{aligned}
\mathbb{E}[\|\boldsymbol{g}^k - \nabla f(\boldsymbol{x}^k)\|_2^2] &\leq 2\mathbb{E}[\|\bar{\mu} - \nabla f(\boldsymbol{x}^k)\|_2^2 + \|\boldsymbol{g}^k - \bar{\mu}\|_2^2] \\
&= 2\mathbb{E}[\|z_k\|_2^2|z_{k-1},\ldots,z_0] + \mathbb{E}[\|\bar{z}_{s_k}\|_2^2|\bar{z}_{s_k-1},\ldots,\bar{z}_0;z_{k-1},\ldots,z_0] \\
&\leq \frac{8G^2}{n - b_k}\left[1 - \frac{(n-b_k)^2 - b_k}{n(n-b_k)}\right] + \frac{8G^2}{s_k}\left[1 - \frac{s_k - 1}{n - b_k}\right].
\end{aligned}$$

This completes the proof. $\qquad\square$

We are now in the position to prove Lemma 1.

*Proof of Lemma 1.* Since we have $n \geq \sum_{i=0}^k s_i$ and $s_k$ is monotone increasing, it follows $n - \sum_{i=0}^{k-1} s_i \geq s_k \geq s_{k-1}$. So in Lemma 3, we have

$$1 - \frac{(n - \sum_{i=0}^{k-1} s_i)^2 - \sum_{i=0}^{k-1} s_i}{n(n - \sum_{i=0}^{k-1} s_i)} \leq 1 + \frac{\sum_{i=0}^{k-1} s_i}{n(n - \sum_{i=0}^{k-1} s_i)} \leq 1 + \frac{1}{n - \sum_{i=0}^{k-1} s_i} \leq 2. \tag{8}$$

Therefore, plugging this into Lemma 3, we can further obtain

$$\mathbb{E}\|\boldsymbol{g}^k - \mu\|_2^2 \leq \frac{24G^2}{s_k}.$$

This proves the desired bound in the lemma. $\qquad\square$

# B  Proofs of Results in Section 3

For brevity, here we use $f^k$ and $f^*$ to denote $f(\boldsymbol{x}^k)$ and $f(\boldsymbol{x}^*)$, respectively.

## B.1  Proof of Theorem 1

Before we prove Theorem 1, we first give a useful corollary derived from Lemma 1.

**Corollary 1.** *Let the sub-sampled gradient $\boldsymbol{g}_k$ be defined in Algorithm 1 without replacement and $s_{k+1} \geq s_k (k \geq 0)$. Then we have*

$$\mathbb{E}[\|\boldsymbol{g}^k - \nabla f(\boldsymbol{x}^k)\|_2] \leq \frac{4G}{\sqrt{s_k}}. \tag{9}$$

*where $G = \max_i \|\nabla f_i - \mu\|_2$.*

*Proof.* From Eqn. (8) in proof of Lemma 1 we have

$$1 - \frac{(n - \sum_{i=0}^{k-1} s_i)^2 - \sum_{i=0}^{k-1} s_i}{n(n - \sum_{i=0}^{k-1} s_i)} \leq 2.$$

Therefore, by using Eqn. (6) and (7) we can further obtain

$$\mathbb{E}\|\boldsymbol{g}^k - \nabla f(\boldsymbol{x}^k)\|_2 \leq \frac{(2\sqrt{2}+1)G}{\sqrt{s_k}} \leq \frac{4G}{\sqrt{s_k}}.$$

The proof is completed. $\qquad\square$

Now we begin to prove Theorem 1. For brevity, here we use $f^k$ and $f^*$ to denote $f(\boldsymbol{x}^k)$ and $f(\boldsymbol{x}^*)$, respectively.

*Proof.* Now we begin to prove the linear convergence of HSGD. We firstly give an useful inequality:

$$
\begin{aligned}
\mathbb{E}\langle \boldsymbol{x}^k - \boldsymbol{x}^* - \eta_k \nabla f^k, \nabla f^k - \boldsymbol{g}^k \rangle =& \mathbb{E}\langle \boldsymbol{x}^k - \boldsymbol{x}^* - \eta_k(\nabla f^k - \nabla f^*), \nabla f^k - \boldsymbol{g}^k \rangle \\
=& \mathbb{E}\|\boldsymbol{x}^k - \boldsymbol{x}^* - \eta_k(\nabla f^k - \nabla f^*)\| \cdot \|\nabla f^k - \boldsymbol{g}^k\| \\
=& \mathbb{E}\left(\|\boldsymbol{x}^k - \boldsymbol{x}^*\| + \eta_k\|\nabla f^k - \nabla f^*\|\right)\|\nabla f^k - \boldsymbol{g}^k\| \\
\overset{①}{\leq}& \left(\|\boldsymbol{x}^k - \boldsymbol{x}^*\| + \eta_k\ell\|\boldsymbol{x}^k - \boldsymbol{x}^*\|\right)\frac{4G}{\sqrt{s_k}} \\
\leq& \frac{4G}{\sqrt{s_k}}\left(1 + \eta_k\ell\right)\|\boldsymbol{x}^k - \boldsymbol{x}^*\|
\end{aligned}
\tag{10}
$$

where ① holds since $f(\boldsymbol{x})$ is $\ell$-smooth and we can bound $\mathbb{E}\|\nabla f^k - \boldsymbol{g}^k\|$ by using Corollary 1.

Then we give the recurrence relation between $\mathbb{E}\|\boldsymbol{x}^{k+1} - \boldsymbol{x}^*\|^2$ and $\mathbb{E}\|\boldsymbol{x}^k - \boldsymbol{x}^*\|^2$ as follows:

$$
\begin{aligned}
&\mathbb{E}\|\boldsymbol{x}^{k+1} - \boldsymbol{x}^*\|^2 \\
=&\mathbb{E}\|\Phi_{\boldsymbol{\mathcal{X}}}(\boldsymbol{x}^k - \boldsymbol{x}^* - \eta_k\boldsymbol{g}^k)\|^2 \\
=&\mathbb{E}\|\boldsymbol{x}^k - \boldsymbol{x}^* - \eta_k\nabla f^k - \eta_k(\nabla f^k - \boldsymbol{g}^k)\|^2 \\
=&\mathbb{E}\left(\|\boldsymbol{x}^k - \boldsymbol{x}^* - \eta_k\nabla f^k\|^2 + \eta_k^2\|\nabla f^k - \boldsymbol{g}^k\|^2 - 2\eta_k\langle \boldsymbol{x}^k - \boldsymbol{x}^* - \eta_k\nabla f^k, \nabla f^k - \boldsymbol{g}^k\rangle\right) \\
=&\mathbb{E}\left(\|\boldsymbol{x}^k - \boldsymbol{x}^*\|^2 - 2\eta_k\langle \boldsymbol{x}^k - \boldsymbol{x}^*, \nabla f^k - \nabla f^*\rangle + \eta_k^2\|\nabla f^k - \nabla f^*\|^2\right) + \eta_k^2\mathbb{E}\|\nabla f^k - \boldsymbol{g}^k\|^2 \\
&- 2\eta_k\mathbb{E}\langle \boldsymbol{x}^k - \boldsymbol{x}^* - \eta_k\nabla f^k, \nabla f^k - \boldsymbol{g}^k\rangle \\
\overset{①}{\leq}&\mathbb{E}\left(\|\boldsymbol{x}^k - \boldsymbol{x}^*\|^2 - 2\eta_k\rho\|\boldsymbol{x}^k - \boldsymbol{x}^*\|^2 + \eta_k^2\ell^2\|\boldsymbol{x}^k - \boldsymbol{x}^*\|^2\right) + \eta_k^2\mathbb{E}\|\nabla f^k - \boldsymbol{g}^k\|^2 \\
&- 2\eta_k\mathbb{E}\langle \boldsymbol{x}^k - \boldsymbol{x}^* - \eta_k\nabla f^k, \nabla f^k - \boldsymbol{g}^k\rangle \\
\overset{②}{\leq}&(1 - 2\eta_k\rho + \eta_k^2\ell^2)\|\boldsymbol{x}^k - \boldsymbol{x}^*\|^2 + \eta_k^2\frac{24G^2}{s_k} + \frac{8G\eta_k}{\sqrt{s_k}}\left(1 + \eta_k\ell\right)\|\boldsymbol{x}^k - \boldsymbol{x}^*\|,
\end{aligned}
$$

where ① holds because we use the $\ell$-smooth property of $f(\boldsymbol{x})$, and for a strong convex function $f(\boldsymbol{x})$, we have the monotonicity of $\nabla f$:

$$\langle \boldsymbol{x}^k - \boldsymbol{x}^*, \nabla f^k - \nabla f^*\rangle \geq \rho\|\boldsymbol{x}^k - \boldsymbol{x}^*\|^2.$$

② holds due to Lemma 1 and Eqn. (10).

Here we set $\eta_k = \frac{\rho}{\ell^2}$ and $s_k = \tau(1/\zeta)^k$, where $\zeta \in (0,1)$. Then consider $\ell \geq \rho$, it yields

$$\mathbb{E}\|\boldsymbol{x}^{k+1} - \boldsymbol{x}^*\|^2 \leq \left(1 - \frac{\rho^2}{\ell^2}\right)\mathbb{E}\|\boldsymbol{x}^k - \boldsymbol{x}^*\|^2 + \frac{8\rho G}{\ell^2\sqrt{\tau}}\left(1 + \frac{\rho}{\ell}\right)\zeta^{k/2}\|\boldsymbol{x}^k - \boldsymbol{x}^*\| + \frac{24\rho^2 G^2}{\tau\ell^4}\zeta^k.$$

For brevity, let $\alpha = 1 - \frac{\rho^2}{\ell^2}$, $\beta = \frac{8\rho G}{\ell^2\sqrt{\tau}}\left(1 + \frac{\rho}{\ell}\right)$ and $\gamma = \frac{24\rho^2 G^2}{\tau\ell^4}$. Thus, we have

$$\mathbb{E}\|\boldsymbol{x}^{k+1} - \boldsymbol{x}^*\|^2 = \alpha\mathbb{E}\|\boldsymbol{x}^k - \boldsymbol{x}^*\|^2 + \beta\zeta^{k/2}\mathbb{E}\|\boldsymbol{x}^k - \boldsymbol{x}^*\| + \gamma\zeta^k.$$

We further assume that $\tau$ is large enough such that

$$\gamma = \frac{24\rho^2 G^2}{\tau \ell^4} \leq \delta \|\boldsymbol{x}^0 - \boldsymbol{x}^*\|^2, \tag{11}$$

where $\delta$ is a positive constant and will be discussed later. Now we use mathematical induction to prove

$$\mathbb{E}\|\boldsymbol{x}^k - \boldsymbol{x}^*\|^2 \leq \theta^k \|\boldsymbol{x}^0 - \boldsymbol{x}^*\|^2 \tag{12}$$

where $\theta < 1$ is a constant and will be given below.

Obviously, when $k = 0$, Eqn. (12) holds. Now assume that for all $t \leq k$, Eqn. (12) holds. Then for $t = k + 1$, we have

$$\begin{aligned}
\mathbb{E}\|\boldsymbol{x}^{k+1} - \boldsymbol{x}^*\|^2 &\leq \alpha \mathbb{E}\|\boldsymbol{x}^k - \boldsymbol{x}^*\|^2 + \beta \zeta^{k/2} \mathbb{E}\|\boldsymbol{x}^k - \boldsymbol{x}^*\| + \gamma \zeta^k \\
&\leq \alpha \theta^k \|\boldsymbol{x}^0 - \boldsymbol{x}^*\|^2 + \beta \zeta^{k/2} \theta^{k/2} \mathbb{E}\|\boldsymbol{x}^0 - \boldsymbol{x}^*\| + \gamma \zeta^k \\
&\overset{①}{\leq} \left( \alpha + \frac{\beta}{\|\boldsymbol{x}^0 - \boldsymbol{x}^*\|} + \delta \right) \theta^k \|\boldsymbol{x}^0 - \boldsymbol{x}^*\|^2 \\
&\overset{②}{\leq} \theta^{k+1} \|\boldsymbol{x}^0 - \boldsymbol{x}^*\|^2,
\end{aligned}$$

where ① and ② hold since we let

$$\theta \geq \max(\zeta, \alpha + \frac{\beta}{\|\boldsymbol{x}^0 - \boldsymbol{x}^*\|} + \delta). \tag{13}$$

This means that if Eqn. (13), then Eqn. (12) always holds. So the conclusion holds.

Now we discuss the values of $\theta$, $\zeta$ and $\tau$ such that Eqn. (12) is satisfied. We just set $\delta = \frac{24\rho^2 G^2}{\tau \ell^4 \|\boldsymbol{x}^0 - \boldsymbol{x}^*\|^2}$, $\tau = \max\left( \frac{324 G^2}{\rho^2 \|\boldsymbol{x}^0 - \boldsymbol{x}^*\|^2}, \frac{432 G^2}{\ell^2 \|\boldsymbol{x}^0 - \boldsymbol{x}^*\|^2} \right)$ and $\theta = \zeta = 1 - \frac{\rho^2}{18\ell^2}$, which gives

$$\begin{aligned}
\theta &\geq \alpha + \frac{\beta}{\|\boldsymbol{x}^0 - \boldsymbol{x}^*\|} + \delta \\
&= 1 - \frac{\rho^2}{\ell^2} + \frac{8\rho G}{\ell^2 \sqrt{\tau}} \left( 1 + \frac{\rho}{\ell} \right) \frac{1}{\|\boldsymbol{x}^0 - \boldsymbol{x}^*\|^2} + \frac{24\rho^2 G^2}{\tau \ell^4 \|\boldsymbol{x}^0 - \boldsymbol{x}^*\|^2} \\
&\geq 1 - \frac{\rho^2}{\ell^2} + \frac{8\rho^2}{9\ell^2} + \frac{\rho^2}{18\ell^2} = 1 - \frac{\rho^2}{18\ell^2}.
\end{aligned}$$

In this case, all the conditions, including Eqn. (11) and (13). So we can see that the values of $\theta$, $\zeta$ and $\tau$ are proper. Therefore, we have

$$\mathbb{E}\|\boldsymbol{x}^k - \boldsymbol{x}^*\|^2 \leq \left( 1 - \frac{\rho^2}{18\ell^2} \right)^k \|\boldsymbol{x}^0 - \boldsymbol{x}^*\|^2.$$

The proof if completed. □

## B.2  Proof of Corollary 1

*Proof.* To achieve $\epsilon$-accurate solution, *i.e.*

$$\|\boldsymbol{x}^k - \boldsymbol{x}^*\|^2 \leq \theta^k \|\boldsymbol{x}^0 - \boldsymbol{x}^*\|^2 \leq \epsilon,$$

where $\theta = 1 - \frac{\rho^2}{18\ell^2}$, we have

$$k^* \geq \log_{1/\theta} \left( \frac{\|\boldsymbol{x}_0 - \boldsymbol{x}^*\|^2}{\epsilon} \right).$$

Therefore, the IFO complexity is

$$\begin{aligned}
\tau \left[ 1 + \frac{1}{\zeta} + \cdots + \frac{1}{\zeta^{k^*-1}} \right] &= \tau \frac{(1/\zeta)^{\log_{1/\theta}\left( \frac{\|\boldsymbol{x}_0 - \boldsymbol{x}^*\|^2}{\epsilon} \right)} - 1}{1/\zeta - 1} = \frac{\tau}{1/\zeta - 1} \left[ \frac{\|\boldsymbol{x}_0 - \boldsymbol{x}^*\|^2}{\epsilon} - 1 \right] \\
&\leq \frac{\tau}{1/\zeta - 1} \left[ \frac{\|\boldsymbol{x}_0 - \boldsymbol{x}^*\|^2}{\epsilon} \right] = \mathcal{O}\left( \frac{\ell^2 G^2}{\rho^2 \epsilon} \right).
\end{aligned}$$

This means that we have the IFO complexity $\mathcal{O}\left( \frac{\ell^2 G^2}{\rho^2 \epsilon} \right)$. The proof is completed. □

## B.3 Proof of Theorem 2

*Proof.* Now we begin to prove the linear convergence of WoRS-based HSGD. Firstly, by smooth property, we have

$$
\begin{aligned}
\mathbb{E} f^{k+1} \leq & \mathbb{E}\left[f^k + \langle \nabla f^k, \boldsymbol{x}^{k+1} - \boldsymbol{x}^k \rangle + \frac{\ell}{2}\|\boldsymbol{x}^{k+1} - \boldsymbol{x}^k\|^2\right] \\
\overset{\text{①}}{=} & \mathbb{E}\left[f^k - \eta_k\langle \nabla f^k, \boldsymbol{g}^k - \nabla f^k + \nabla f^k \rangle + \frac{\ell}{2}\|\Phi_{\boldsymbol{\mathcal{X}}}\left(\boldsymbol{x}^k - \eta_k \boldsymbol{g}^k\right) - \boldsymbol{x}^k\|^2\right] \\
= & \mathbb{E}\left[f^k - \eta_k\langle \nabla f^k, \boldsymbol{g}^k - \nabla f^k + \nabla f^k \rangle + \frac{\ell}{2}\|\Phi_{\boldsymbol{\mathcal{X}}}\left(\eta_k \boldsymbol{g}^k\right)\|^2\right] \\
\leq & \mathbb{E}\left[f^k - \eta_k\langle \nabla f^k, \boldsymbol{g}^k - \nabla f^k + \nabla f^k \rangle + \frac{\ell\eta_k^2}{2}\|\boldsymbol{g}^k - \nabla f^k + \nabla f^k\|^2\right] \\
= & \mathbb{E}\left[f^k - \eta_k(1 - \eta_k\ell)\langle \nabla f^k, \boldsymbol{g}^k - \nabla f^k \rangle + \frac{\ell\eta_k^2}{2}\|\boldsymbol{g}^k - \nabla f^k\|^2 - \eta_k\left(1 - \frac{\ell\eta_k}{2}\right)\|\nabla f^k\|^2\right],
\end{aligned}
$$

where ① holds due to $\boldsymbol{x}^k \in \boldsymbol{\mathcal{X}}$. Here we set $\eta_k = \frac{1}{\ell}$ and plug it into the above inequality:

$$
\mathbb{E} f^{k+1} \leq \mathbb{E}\left[f^k + \frac{1}{2\ell}\|\boldsymbol{g}^k - \nabla f^k\|^2 - \frac{1}{2\ell}\|\nabla f^k\|^2\right] \overset{\text{①}}{\leq} \mathbb{E}\left[f^k + \frac{12G^2}{\ell s_k} - \frac{1}{2\ell}\|\nabla f^k\|^2\right], \quad (14)
$$

where ① holds since we can bound $\mathbb{E}\|\nabla f^k - \boldsymbol{g}^k\|_2^2$ by using Lemma 1.

On the other hand, $f(\boldsymbol{x})$ is a strongly convex function. Namely, we have

$$
f(\boldsymbol{y}) \geq f(\boldsymbol{x}) + \nabla f(\boldsymbol{x})^T(\boldsymbol{y} - \boldsymbol{x}) + \frac{\rho}{2}\|\boldsymbol{y} - \boldsymbol{x}\|^2.
$$

Then by minimizing $\boldsymbol{y}$ on both sides, it yields

$$
\frac{1}{2\rho}\|\nabla f(\boldsymbol{x})\|^2 \geq f(\boldsymbol{x}) - f(\boldsymbol{x}^*). \quad (15)
$$

We plug Eqn. (15) into Eqn. (14) and obtain

$$
\mathbb{E}(f^{k+1} - f^*) \leq \left(1 - \frac{\rho}{\ell}\right)(f^k - f^*) + \frac{12G^2}{\ell s_k}.
$$

Here we set $s_k = \tau(1/\zeta)^k$, where $\zeta \in (0, 1)$. For brevity, let $\alpha = 1 - \frac{\rho}{\ell}$ and $\gamma = \frac{12G^2}{\tau\ell}$. It yields

$$
\mathbb{E}(f^{k+1} - f^*) \leq \alpha(f^k - f^*) + \gamma\zeta^k.
$$

We further assume that $\tau$ is large enough such that

$$
\gamma = \frac{12G^2}{\tau\ell} \leq \delta(f^0 - f^*), \quad (16)
$$

where $\delta$ is a positive constant and will be discussed later. Now we use mathematical induction to prove

$$
f^k - f^* \leq \theta^k(f^0 - f^*), \ (\forall k), \quad (17)
$$

where $\theta < 1$ is a constant and will be given below.

Obviously, when $k = 0$, Eqn. (17) holds. Now assume that for all $t \leq k$, Eqn. (17) holds. Then for $t = k + 1$, we have

$$
\begin{aligned}
\mathbb{E}(f^{k+1} - f^*) \leq & \alpha\mathbb{E}(f^k - f^*) + \gamma\zeta^k \leq \alpha\theta^k(f^0 - f^*) + \gamma\zeta^k \\
\overset{\text{①}}{\leq} & (\alpha + \delta)\theta^k(f^0 - f^*) \overset{\text{②}}{\leq} \theta^{k+1}(f^0 - f^*),
\end{aligned}
$$

where ① and ② hold since we let

$$
\theta \geq \max(\zeta, \alpha + \delta). \quad (18)
$$

This means that if Eqn. (18) holds, then Eqn. (17) always holds. So the conclusion holds.

Now we discuss the values of $\theta$, $\zeta$ and $\tau$ such that Eqn. (18) is satisfied. We just set $\delta = \frac{12G^2}{\tau\ell(f^0-f^*)}$, $\tau \geq \frac{6G^2}{\rho(f^0-f^*)}$ and $\theta = \zeta = 1 - \frac{\rho}{2\ell}$, giving

$$\theta \geq \alpha + \delta \geq 1 - \frac{\rho}{\ell} + \frac{\rho}{2\ell} = 1 - \frac{\rho}{2\ell}.$$

In this case, all the conditions hold, including Eqn. (16) and (18). So we can see that the values of $\theta$, $\zeta$ and $\tau$ are proper. Therefore, we have

$$\mathbb{E}(f^k - f^*) \leq \left(1 - \frac{\rho}{2\ell}\right)^k (f^0 - f^*).$$

Then we derive the IFO complexity. To achieve $\epsilon$-accurate solution, *i.e.*

$$\mathbb{E}(f^k - f^*) \leq \theta^k(f^0 - f^*) \leq \epsilon,$$

where $\theta = 1 - \frac{\rho}{2\ell}$, we have

$$k^* \geq \log_{1/\theta}\left(\frac{f^0 - f^*}{\epsilon}\right).$$

Therefore, the IFO complexity is

$$\tau\left[1 + \frac{1}{\zeta} + \cdots + \frac{1}{\zeta^{k^*-1}}\right] = \tau\frac{(1/\zeta)^{\log_{1/\theta}\left(\frac{f^0-f^*}{\epsilon}\right)} - 1}{1/\zeta - 1} = \frac{\tau}{1/\zeta - 1}\left[\frac{f^0 - f^*}{\epsilon} - 1\right]$$
$$\leq \frac{\tau}{1/\zeta - 1}\left[\frac{f^0 - f^*}{\epsilon}\right] \leq \mathcal{O}\left(\frac{\kappa G^2}{\epsilon}\right),$$

where $\kappa = \ell/\rho$. This means that we have the IFO complexity $\mathcal{O}\left(\frac{\kappa G^2}{\epsilon}\right)$. The proof is completed.

The proof is completed.

$\square$

## B.4 Proof of Theorem 3

For brevity, here we use $f^k$ and $f^*$ to denote $f(\boldsymbol{x}^k)$ and $f(\boldsymbol{x}^*)$, respectively.

*Proof.* From Eqn. (10) in Appendix B.1, we have

$$\mathbb{E}\langle\boldsymbol{x}^k - \boldsymbol{x}^* - \eta_k\nabla f^k, \nabla f^k - \boldsymbol{g}^k\rangle \leq \frac{4G}{\sqrt{s_k}}(1 + \eta_k\ell)\|\boldsymbol{x}^k - \boldsymbol{x}^*\|.$$

For arbitrary $\boldsymbol{x}_1 \in \mathcal{X}$ and $\boldsymbol{x}_2 \in \mathcal{X}$ that satisfy $\|\boldsymbol{x}_1 - \boldsymbol{x}_2\|_2 \leq D$, we can bound $\mathbb{E}\langle\boldsymbol{x}^k - \boldsymbol{x}^* - \eta_k\nabla f^k, \nabla f^k - \boldsymbol{g}^k\rangle$ as follows:

$$\mathbb{E}\langle\boldsymbol{x}^k - \boldsymbol{x}^* - \eta_k\nabla f^k, \nabla f^k - \boldsymbol{g}^k\rangle \overset{①}{\leq} \frac{4G}{\sqrt{s_k}}(1 + \eta_k\ell)\|\boldsymbol{x}^k - \boldsymbol{x}^*\| \leq \frac{4(1 + \eta_k\ell)GD}{\sqrt{s_k}}. \qquad (19)$$

Then we utilize Eqn. (19) to further give the relationship between $\mathbb{E}\|\boldsymbol{x}^{k+1} - \boldsymbol{x}^*\|^2$ and $\mathbb{E}\|\boldsymbol{x}^k - \boldsymbol{x}^*\|^2$:

$$
\begin{aligned}
&\mathbb{E}\|\boldsymbol{x}^{k+1} - \boldsymbol{x}^*\|^2 \\
=&\mathbb{E}\|\Phi_{\boldsymbol{\mathcal{X}}}\left(\boldsymbol{x}^k - \eta_k \boldsymbol{g}^k\right) - \boldsymbol{x}^*\|^2 \\
\overset{\text{\textcircled{1}}}{\leq}&\mathbb{E}\|\boldsymbol{x}^k - \boldsymbol{x}^* - \eta_k \nabla f^k - \eta_k(\nabla f^k - \boldsymbol{g}^k)\|^2 \\
=&\mathbb{E}\left(\|\boldsymbol{x}^k - \boldsymbol{x}^* - \eta_k \nabla f^k\|^2 + \eta_k^2\|\nabla f^k - \boldsymbol{g}^k\|^2 - 2\eta_k\langle\boldsymbol{x}^k - \boldsymbol{x}^* - \eta_k \nabla f^k, \nabla f^k - \boldsymbol{g}^k\rangle\right) \\
\overset{\text{\textcircled{2}}}{\leq}&\mathbb{E}\|\boldsymbol{x}^k - \boldsymbol{x}^* - \eta_k \nabla f^k\|^2 + \frac{8\eta_k(1 + \eta_k\ell)GD}{\sqrt{s_k}} + \frac{24\eta_k^2 G^2}{s_k} \\
=&\mathbb{E}\left(\|\boldsymbol{x}^k - \boldsymbol{x}^*\|^2 - 2\eta_k\langle\boldsymbol{x}^k - \boldsymbol{x}^*, \nabla f^k\rangle + \eta_k^2\|\nabla f^k\|^2\right) + \frac{8\eta_k(1 + \eta_k\ell)GD}{\sqrt{s_k}} + \frac{24\eta_k^2 G^2}{s_k} \\
\overset{\text{\textcircled{3}}}{\leq}&\mathbb{E}\left(\|\boldsymbol{x}^k - \boldsymbol{x}^*\|^2 + 2\eta_k(f^* - f^k) + 2\ell\eta_k^2(f^k - f^*)\right) + \frac{8\eta_k(1 + \eta_k\ell)GD}{\sqrt{s_k}} + \frac{24\eta_k^2 G^2}{s_k} \\
=&\mathbb{E}\left(\|\boldsymbol{x}^k - \boldsymbol{x}^*\|^2 - 2\eta_k(1 - \ell\eta_k)(f^k - f^*)\right) + \frac{8\eta_k(1 + \eta_k\ell)GD}{\sqrt{s_k}} + \frac{24\eta_k^2 G^2}{s_k},
\end{aligned}
\tag{20}
$$

where ① holds due to $\boldsymbol{x}^* \in \boldsymbol{\mathcal{X}}$. ② holds since we use Corollary 1 and Eqn. (19), and ③ holds due to the convexity of $f(\boldsymbol{x})$:

$$
f^* - f^k \geq -\langle\nabla f^k, \boldsymbol{x}^k - \boldsymbol{x}^*\rangle,
$$

and the $\ell$-smooth property of $f(\boldsymbol{x})$:

$$
f^* \leq \inf_{\boldsymbol{y}}\left(f(\boldsymbol{x}) + \langle\nabla f(\boldsymbol{x}), \boldsymbol{y} - \boldsymbol{x}\rangle + \frac{\ell}{2}\|\boldsymbol{y} - \boldsymbol{x}\|^2\right) = f(\boldsymbol{x}) - \frac{1}{2\ell}\|\nabla f(\boldsymbol{x})\|^2,
$$

where we set $\boldsymbol{y} = \boldsymbol{x} - \nabla f(\boldsymbol{x})/\ell$.

Next we sum up Eqn. (20) from $k = \theta T$ to $T - 1$ and obtain

$$
\sum_{k=\theta T}^{T-1} 2\eta_k(1 - \ell\eta_k)\mathbb{E}(f^k - f^*) \leq \|\boldsymbol{x}^{\theta T} - \boldsymbol{x}^*\|^2 - \|\boldsymbol{x}^T - \boldsymbol{x}^*\|^2 + \sum_{k=\theta T}^{T-1}\left[\frac{8\eta_k(1 + \eta_k\ell)GD}{\sqrt{s_k}} + \frac{24\eta_k^2 G^2}{s_k}\right].
$$

Here we set $\eta_k = \frac{1}{2\ell}$. Then it yields

$$
\begin{aligned}
&\frac{1}{(1 - \theta)T}\sum_{k=\theta T}^{T-1}\mathbb{E}(f^k - f^*) \\
\leq&\frac{2\ell}{(1 - \theta)T}\left(\|\boldsymbol{x}^{\theta T} - \boldsymbol{x}^*\|^2 - \|\boldsymbol{x}^T - \boldsymbol{x}^*\|^2\right) + \frac{1}{(1 - \theta)T}\sum_{k=\theta T}^{T-1}\left(\frac{12GD}{\sqrt{s_k}} + \frac{12G^2}{\ell s_k}\right).
\end{aligned}
\tag{21}
$$

Then, we further set $s_k = (k + 1)^2$. We have

$$
\sum_{k=\theta T}^{T-1}\frac{1}{\sqrt{s_k}} = \sum_{k=\theta T}^{T-1}\frac{1}{k + 1} \leq \int_{\theta T}^{T-1}\frac{1}{x}dx = \log(x)\big|_{\theta T}^{T-1} \leq \log\left(\frac{1}{\theta}\right)
$$

and

$$
\sum_{k=\theta T}^{T-1}\frac{1}{s_k} = \sum_{k=\theta T}^{T-1}\frac{1}{(k + 1)^2} \leq \sum_{k=\theta T}^{T-1}\left(\frac{1}{k} - \frac{1}{k + 1}\right) \leq \frac{1}{\theta T}.
$$

Finally, we submit the above inequalities into Eqn. (21) and set $\theta = \frac{1}{2}$:

$$
\begin{aligned}
\mathbb{E}(f(\boldsymbol{x}^a) - f(\boldsymbol{x}^*)) =&\frac{1}{(1 - \theta)T}\sum_{k=\theta T}^{T-1}\mathbb{E}(f^k - f^*) \\
\leq&\frac{4\ell}{T}\|\boldsymbol{x}^{\theta T} - \boldsymbol{x}^*\|^2 + \frac{24GD}{T} + \frac{48G^2}{\ell T^2} \leq \frac{4\ell D^2 + 24GD}{T} + \frac{48G^2}{\ell T^2}.
\end{aligned}
$$

The proof is completed. $\qquad\square$

## B.5 Proof of Corollary 2

*Proof.* From Theorem 3, we know that the convergence rate is decided by $\mathcal{O}((6GD + \ell D^2)/T)$. In order to achieve $\epsilon$ accuracy, we need $T \geq \mathcal{O}(\frac{6GD + \ell D^2}{\epsilon})$. So the IFO complexity of the algorithm is

$$\mathcal{O}(1^2 + 2^2 + \cdots + T^2) = \mathcal{O}\left(\frac{(6GD + \ell D^2)^3}{\epsilon^3}\right).$$

The proof is completed. $\square$

## B.6 Proof of Theorem 4

For brevity, here we use $f^k$ and $f^*$ to denote $f(\boldsymbol{x}^k)$ and $f(\boldsymbol{x}^*)$, respectively.

*Proof.* From Eqn. (14) in Sec. B.3, by setting $\eta_k = \frac{1}{\ell}$ we have

$$\mathbb{E}f^{k+1} \leq \mathbb{E}\left[f^k + \frac{12G^2}{\ell s_k} - \frac{1}{2\ell}\|\nabla f^k\|^2\right]. \tag{22}$$

We set $s_k = k + 1$ and sum up Eqn. (22) from $k = \theta T$ to $T - 1$:

$$\frac{1}{(1-\theta)T}\sum_{k=\theta T}^{T-1}\mathbb{E}\|\nabla f^k\|^2 \leq \frac{2\ell}{(1-\theta)T}(f^{\theta T} - f^T) + \frac{24G^2}{(1-\theta)T}\sum_{k=\theta T}^{T-1}\frac{1}{s_k}$$

$$\overset{\textcircled{1}}{\leq} \frac{2\ell}{(1-\theta)T}(f^{\theta T} - f^T) + \frac{24G^2}{(1-\theta)T}\log\left(\frac{1}{\theta}\right),$$

where $\textcircled{1}$ holds since we have

$$\sum_{k=\theta_1 T + 1}^{\theta_2 T}\frac{1}{s_k} \leq \int_{\theta_1 T}^{\theta_2 T - 1}\frac{1}{x}dx = \log(x)\Big|_{\theta_1 T}^{\theta_2 T - 1} \leq \log\left(\frac{\theta_2}{\theta_1}\right).$$

Suppose we are given arbitrary $\boldsymbol{x}_1 \in \mathcal{X}$ and $\boldsymbol{x}_2 \in \mathcal{X}$ that satisfy $\|\boldsymbol{x}_1 - \boldsymbol{x}_2\|_2 \leq D$, and $f(\boldsymbol{x})$ is $\ell$-smooth. We have

$$f^{\theta T} - f^T = (f^{\theta T} - f^*) - (f^T - f^*) \leq \frac{\ell}{2}\|\boldsymbol{x}^{\theta T} - \boldsymbol{x}^*\|_2^2 + \frac{\ell}{2}\|\boldsymbol{x}^T - \boldsymbol{x}^*\|_2^2 \leq \ell D^2.$$

By setting $\theta = 1/2$, we can further establish

$$\mathbb{E}\|\nabla f(\boldsymbol{x}^a)\|^2 = \frac{1}{0.5T}\sum_{k=0.5T+1}^{T}\mathbb{E}\|\nabla f^k\|^2 \leq \frac{4\ell^2 D^2 + 35G^2}{T}.$$

The proof if completed. $\square$

## B.7 Proof of Corollary 3

*Proof.* From Theorem 4, we know

$$\mathbb{E}\|\nabla f(\boldsymbol{x}^a)\|^2 \leq \frac{4\ell^2 D^2 + 35G^2}{T}.$$

In this case, we can further achieve $\mathbb{E}\|\nabla f(\boldsymbol{x}^a)\|^2 \leq \epsilon$. We need $T \geq \frac{4\ell^2 D^2 + 35G^2}{\epsilon}$. So the IFO complexity is

$$\mathcal{O}\left(\frac{\left(4\ell^2 D^2 + 35G^2\right)^2}{\epsilon^2}\right).$$

The proof is completed. $\square$

# C  Proofs of Results in Section 4

## C.1  Proof of Theorem 5

Before proving Theorem 5, we first give a useful lemma stated in Lemma 4.

**Lemma 4.** *[1] For the convex function $f(\boldsymbol{x}) = g(\boldsymbol{Ax})$, if the function $g(\cdot)$ is $\alpha$-strongly convex and $\mathcal{X}$ is a compact set, then $f(\boldsymbol{x})$ satisfies Polyak-Łojasiewicz (PL) inequality:*

$$\mu(f(\boldsymbol{x}) - f(\boldsymbol{x}^*)) \leq \frac{1}{2}\|\nabla f(\boldsymbol{x})\|_2^2,$$

*where $\mu = \alpha\sigma(\boldsymbol{A})$ in which $\alpha$ is a universal constant and $\sigma(\boldsymbol{A})$ denotes the smallest non-zero singular value of the matrix $\boldsymbol{A}$.*

Now we are to prove Theorem 5. For brevity, here we use $f^k$ and $f^*$ to denote $f(\boldsymbol{x}^k)$ and $f(\boldsymbol{x}^*)$, respectively.

*Proof.* From Eqn. (14) in Sec. B.3, by setting $\eta_k = \frac{1}{\ell}$ we have

$$\mathbb{E}f^{k+1} \leq \mathbb{E}\left[f^k + \frac{12G^2}{\ell s_k} - \frac{1}{2\ell}\|\nabla f^k\|^2\right]. \tag{23}$$

Then since each individual function $f_i(\boldsymbol{x})$ is of form $f_i(\boldsymbol{x}) = h(\langle \boldsymbol{a}_i, \boldsymbol{x}\rangle)$, we can formulate $f(\boldsymbol{x}) = g(\boldsymbol{Ax})$, where $\boldsymbol{A} = [\boldsymbol{a}_1^T; \boldsymbol{a}_2^T; \cdots, \boldsymbol{a}_n^T]$ (namely, each row denotes a datum vector). Since $h(\cdot)$ is stongly convex, then by Lemma 4 we know $g'(\boldsymbol{x}) = g(\boldsymbol{Ax})$ satisfies the Polyak-Łojasiewicz (PL) inequality:

$$\mu(g'(\boldsymbol{x}) - g'(\boldsymbol{x}^*)) \leq \frac{1}{2}\|\nabla g'(\boldsymbol{x})\|_2^2,$$

where $\mu = \alpha\sigma(\boldsymbol{A})$ in which $\sigma(\boldsymbol{A})$ denotes the smallest non-zero singular value of the matrix $\boldsymbol{A}$. It can be easily verified that $\mu = \alpha\sigma(\boldsymbol{A}) \leq \ell$. Note that the most commonly used optimization losses, namely least square and logistic regression, satisfy such a PL inequality [1]. Thus, by substituting the above PL inequality into Eqn. (23), it yields

$$\mathbb{E}f^{k+1} \leq \mathbb{E}\left[f^k + \frac{12G^2}{\ell s_k} - \frac{\mu}{2\ell}(f^k - f^*)\right],$$

which is actually equivalent to

$$\mathbb{E}[f^{k+1} - f^*] \leq \left(1 - \frac{\mu}{\ell}\right)\mathbb{E}[f^k - f^*] + \frac{12G^2}{\ell s_k}.$$

Then we set $s_k = \tau(1/\zeta)^k$, where $\zeta \in (0, 1)$. Then by considering $\ell \geq \rho$, it yields

$$\mathbb{E}[f^{k+1} - f^*] \leq \left(1 - \frac{\mu}{\ell}\right)\mathbb{E}[f^k - f^*] + \frac{12G^2}{\ell\tau}\zeta^k.$$

For brevity, let $\alpha = 1 - \frac{\mu}{\ell}$ and $\gamma = \frac{12G^2}{\tau\ell}$. Thus, we have

$$\mathbb{E}[f^{k+1} - f^*] \leq \alpha[f^k - f^*] + \gamma\zeta^k.$$

We further assume that $\tau$ is large enough such that

$$\gamma = \frac{12G^2}{\tau\ell} \leq \delta(f^0 - f^*), \tag{24}$$

where $\delta$ is a positive constant and will be discussed later. Now we use mathematical induction to prove

$$\mathbb{E}(f^k - f^*) \leq \theta^k \left(f^0 - f^*\right), \tag{25}$$

where $\theta < 1$ is a constant and will be given below.

Obviously, when $k = 0$, Eqn. (25) holds. Now assume that for all $t \leq k$, Eqn. (25) holds. Then for $t = k + 1$, we have

$$
\begin{aligned}
\mathbb{E}(f^{k+1} - f^*) &\leq \alpha \mathbb{E}(f^k - f^*) + \beta \zeta^k \\
&\leq \alpha \theta^k \left( f^0 - f^* \right) + \beta \zeta^k \\
&\stackrel{\text{①}}{\leq} (\alpha + \delta) \theta^k \left( f^0 - f^* \right) \\
&\stackrel{\text{②}}{\leq} \theta^{k+1} \left( f^0 - f^* \right),
\end{aligned}
$$

where ① and ② hold since we let

$$
\theta \geq \max(\zeta, \alpha + \delta). \tag{26}
$$

This means that if Eqn. (26) holds, then Eqn. (25) always holds. So the conclusion holds.

Now we discuss the values of $\theta$, $\zeta$ and $\tau$ to make Eqn. (26) satisfied. We just set $\delta = \frac{\mu}{2\ell}$, $\tau \geq \frac{24G^2}{\mu(f^0 - f^*)}$ and $\theta = \zeta = 1 - \frac{\mu}{2\ell}$, giving

$$
\theta \geq \alpha + \delta = 1 - \frac{\mu}{2\ell}.
$$

In this case, all the conditions hold, including Eqn. (24) and (26). So we can see that the values of $\theta$, $\zeta$ and $\tau$ are proper. Therefore, we have

$$
\mathbb{E}(f^k - f^*) \leq \left( 1 - \frac{\mu}{2\ell} \right)^k (f^0 - f^*).
$$

The proof if completed. $\qquad\square$

## C.2 Proof of Corollary 4

*Proof.* To achieve $\epsilon$-accurate solution, *i.e.*

$$
\mathbb{E}(f^k - f^*) \leq \theta^k (f^0 - f^*) \leq \epsilon,
$$

where $\theta = 1 - \frac{\mu}{2\ell}$, we have

$$
k^* \geq \log_{1/\theta} \left( \frac{f^0 - f^*}{\epsilon} \right).
$$

Therefore, the IFO complexity is

$$
\begin{aligned}
\tau \left[ 1 + \frac{1}{\zeta} + \cdots + \frac{1}{\zeta^{k^*-1}} \right] &= \tau \frac{(1/\zeta)^{\log_{1/\theta}\left( \frac{f^0 - f^*}{\epsilon} \right)} - 1}{1/\zeta - 1} = \frac{\tau}{1/\zeta - 1} \left[ \frac{f^0 - f^*}{\epsilon} - 1 \right] \\
&\leq \frac{\tau}{1/\zeta - 1} \left[ \frac{f^0 - f^*}{\epsilon} \right] \leq \mathcal{O}\left( \frac{48\ell G^2}{\mu^2 \epsilon} \right).
\end{aligned}
$$

The proof is completed. $\qquad\square$

# D  Additional Experimental Results

## D.1  Descriptions of Testing Datasets and Compared Algorithms

We first briefly introduce the ten testing datasets in the manuscript. Among them, nine datasets are provided in the LibSVM website[1], including ijcnn1, a9a, w8a, covtype, rcv11, protein, satimage, sensorless and letter. We also evaluate our algorithms on the mnist[2] dataset, which is a very commonly used handwritting recognition dataset. Their detailed information is summarized in Table 2. We can observe that these datasets are different from each other in feature dimension, training samples, and class numbers, *etc*.

Now we briefly introduce the compared algorithms in the manuscript, including SVRG [2], SAGA [3], AVRG [4] and SCGC [5]. Since SGD is well known, here we do not introduce it.

Table 2: Descriptions of the ten testing datasets.

|  | #class | #sample | #feature |  | #class | #sample | #feature |
|---|---|---|---|---|---|---|---|
| ijcnn1 | 2 | 49,990 | 22 | protein | 3 | 14,895 | 357 |
| a9a | 2 | 32,561 | 123 | satimage | 6 | 4,435 | 36 |
| w8a | 2 | 49,749 | 300 | sensorless | 7 | 2,310 | 19 |
| covtype | 2 | 581,012 | 54 | letter | 26 | 10,500 | 16 |
| rcv11 | 2 | 20,242 | 47,236 | mnist | 10 | 60,000 | 784 |

- SVRG: It is a variance-reduced variant of SGD. At the $k$-th epoch, it firstly computes the full gradient $\nabla f(\tilde{x})$ at a snapshot point $\tilde{x}$. Typically, the snapshot point $\tilde{x}$ is set to the final output $x^{k-1}$ of the previous epoch. Then it updates the variables as $x_t^k = x_{t-1}^k - \eta_t \left( f_{i_t}(x_{t-1}^k) - f_{i_t}(\tilde{x}) + \nabla f(\tilde{x}) \right)$ where $i_t$ is the sampled index at the $t$-th iteration in the $k$-th epoch. The iteration number $T$ in each epoch is usually set to the sample number $n$ and the final output of the $k$-th epoch is usually the final computed solution $x_T^k$ in this epoch in implementation.

- SAGA: It needs a table to store the gradient of historical computed variables. Specifically, let the initial point denoted by $x^0$ and the known gradient $\nabla f_i(\phi_i^0)$ $(i = 1, \cdots, n)$ where $\phi_i^0 = x^0$. Then at the $k$ iteration, it picks an index $j$ at random. Then it sets $\phi_j^k = x^{k-1}$ and stores $\nabla f_j(\phi_j^k)$ in the table. All other entries in the table remain unchanged. Finally, it updates $x^k$ as $x^k = x^{k-1} - \eta_k \left( f_j(\phi_j^k) - f_j(\phi_j^{k-1}) + \frac{1}{n} \sum_{i=1}^n f_i(\phi_i^{k-1}) \right)$. SAGA is also a variance-reduced method.

- AVRG: It uses the historical gradient to estimate full gradient of the snapshot point in SVRG. Namely, at each epoch, it sums up the gradient $f_{i_t}(x_{t-1}^k)$ and uses its average as the estimation of $f(\tilde{x})$ in next epoch. Such a strategy can reduce computational complexity.

- SCGC: It has similar updating process as SVRG. Namely, it also takes the output in the previous epoch as the current snapshot point. But at each epoch, it only samples a subset $\mathcal{S}_k$ of data and uses the average gradient $g(\tilde{x})$ of the samples in $\mathcal{S}_k$ at the snapshot point $\tilde{x}$ to estimate the full gradient at $\tilde{x}$. During the iteration, it updates $x_t^k$ as $x_t^k = x_{t-1}^k - \eta_t \left( f_{i_t}(x_{t-1}^k) - f_{i_t}(\tilde{x}) + g(\tilde{x}) \right)$ where $i_t$ is the sampled index from $\mathcal{S}_k$ in the $t$-th iteration. Typically, the size of $\mathcal{S}_k$ gradually increases along with more iterations. Note that to date there has been no work analyzing the convergence performance of SCGC under WoRS.

(a) ijcnn     (b) covtype     (c) w8a

(d) rcv11     (e) protein     (f) satimage

Figure 5: WoRS *vs.* WRS in HSGD. We test logistic regression (regularization parameter $\lambda = 0.01$) on ijcnn, covtype, w8a and rcv11, and evaluate softmax regression (regularization parameter $\lambda = 0.1$) on protein and satimage.

## D.2 Comparison between WoRS and WRS in HSGD

Then we present more experimental results to compare WoRS and WRS. Since the $\ell_2$-regularized logistic and multi-class softmax regression problems are strongly convex, we follow Theorem 2 to exponentially expand the mini-batch size $s_k$ in HSGD and set $\tau = 1$. From the comparison in Figure 5, we can find that WoRS strategy often outperforms WRS in the anaphasis of going through data for one pass, while at the beginning of the iteration, their performance is mostly the same. This is because at the beginning, only a few samples are selected and it is highly probable for WRS to select different samples, which is almost the same as WoRS. Thus, their performance in the early phase is very similar. In contrast, as the iteration proceeds, more samples are required. It is likely that WRS selects repeated samples which provide redundant descending information (gradient). By comparison, WoRS has no such weakness as it uses different samples. So it can utilize all samples more effectively and runs faster.

## Footnotes

[1]https://www.csie.ntu.edu.tw/ cjlin/libsvmtools/datasets/

[2]http://yann.lecun.com/exdb/mnist/