[Reviews · NeurIPS 2018]

Reviewer 1



The paper presents a fairly comprehensive analysis of HSGD in a wide variety of settings from strongly convex to non-convex problems under the assumption of sampling without replacement. One of the main challenges in sampling without replacement is to tackle the lack of independence this creates. This is perhaps the main technical contribution of the paper where a comparison process is introduced to enable martingale based arguments. The intuitive description of this in lines 130-136 is quite confusing and one really needs to look at the technical details to understand it. I think easier proofs could be obtained by using the fact that sampling without replacement results in negatively dependent variables for which the usual concentration bounds for independent variables apply unchanged.

Reviewer 2



The article studies the problem of hybrid SGD under without replacement conditions. Theoretical results in terms of convergence are provided in different scenarios. These results are further reinforced with experimental results. The results suggest improvement of the proposed method compared to the rest. However, when checking the figures of convergence we can easily see how the proposed method clearly converges faster and seems to reach a "plateau" stage while some of the rest of the methods still clearly show a decreasing behavior. I was wondering if the authors checked the convergence beyond the IFO/n values shown. Regarding the rest of the paper, the organization is correct but difficult to follow. Mostly because there are many different messages involved. Otherwise a nice paper.

Reviewer 3



This paper lies in the well-developed field of variance-reduced stochastic gradient algorithms. It proposes a theoretical study of the well-known HSGD for sampling without replacement scheme, which is known to perform better than sampling with replacement in practice. The considered setting makes the analysis of the algorithm more difficult, since in this case the mini-batch gradients are not unbiased anymore. Rates for strongly-convex, non-strongly convex and non-convex objectives are provided, with a special care for linearly structured problems (such as generalized linear models). Numerical experiments are convincing enough, although the datasets used in experiments are not very large (largest one is covtype) (while authors argue that this methods is interesting in the large n medium precision setting). I think that this is an nice piece of work, which provides an incremental contribution to the already widely developed theory of stochastic optimisation algorithms for ML. The paper can certainly be accepted for publication, although I think that the contribution is mostly incremental, since (to the best of my knowledge) the proofs do not seem to introduce a major novelty.